



# Contributions to uncertainty related to hydrostratigraphic modeling using Multiple-Point Statistics

Adrian A.S. Barfod[1,2], Troels N. Vilhelmsen[2], Flemming Jørgensen[1], Anders V. Christiansen[2], Julien Straubhaar[3], Ingelise Møller[1]

[1]Derpartment of Groundwater and Quaternary Geological Mapping, Geological Survey of Denmark & Greenland (GEUS), C.F. Møllers Allé 8, 8000 Aarhus C, Denmark
[2]Hydrogeophysics Group, Department of Geoscience, Aarhus University, C.F. Møllers Allé 4, 8000 Aarhus C, Denmark
[3]Centre d'Hydrogéologie et de Géothermie (CHYN), Université de Neuchâtel, Switzerland

*Correspondence to*: Adrian A.S. Barfod (adrian.s.barfod@gmail.com)





# Contents



**Abstract.**

Forecasting the flow of groundwater requires a hydrostratigraphic model, which describes the architecture of the subsurface. State-of-the-art Multiple-Point Statistical (MPS) tools are readily available for creating models depicting subsurface geology. We present a study of the impact of key parameters related to stochastic MPS simulation of a real-world hydrogeophysical dataset from Kasted Denmark using the *snesim* algorithm. The goal is to study how changes to the underlying datasets propagate into the hydrostratigraphic realizations when using MPS for stochastic modeling. This study focuses on the sensitivity of the MPS realizations to the geophysical soft data, borehole lithology logs, and the Training Image (TI). The modeling approach used in this paper utilizes a cognitive geological model as a TI to simulate ensemble hydrostratigraphic models. The target model contains three overall hydrostratigraphic categories, and the MPS realizations are compared visually, as well as quantitatively using mathematical measures of similarity. The quantitative similarity analysis is carried out exhaustively, and realizations are compared with each other as well as with the cognitive geological model.

The results underline the importance of geophysical data for constraining MPS simulations. Relying only on borehole data and the conceptual geology or TI, results in a significant increase in realization uncertainty. The SkyTEM data used in this study cover a large portion of the Kasted model area, and are essential to the hydrostratigraphic architecture. On the other hand, the borehole lithology logs are sparse, and only 410 boreholes were present in this study. The borehole lithology logs infer local changes in the immediate vicinity of the boreholes, thus providing limited large-scale structural information. Lithological information is, however, important for the interpretation of the geophysical responses. Finally, the importance of the TI was studied. An example was presented where an alternative geological model from a neighboring area was used to simulate hydrostratigraphic models. It was shown that as long as the geological setting are similar in nature, the realizations, although different, still reflect the hydrostratigraphic architecture. If a TI containing a biased geological conceptualization is used, the resulting realizations will resemble the TI and contain less structure in particular areas, where the soft data show almost even probability to two or all three of the hydrostratigraphic units.

# 1 Introduction

Geological models are important from both a societal and economic perspective, since they are used to locate essential natural resources, such as freshwater, oil, metals, rare earth minerals *etc*. Additionally, they are used in risk assessment related to natural hazards, such as: earthquakes, sinkholes, volcanic eruptions, and landslides. Building 3D models depicting real-world subsurface geology is no trivial task. Information from multiple sources is required, *i.e.* conceptual geological understanding, geological information, lithology logs and geophysical data. Such data are sparse, uncertain and redundant. Dataset gaps force geoscientists to make uncertain predictions or estimates, which carries over into the resulting geological model. During the modeling procedure, such problems are dealt with as best as possible. Gaps in knowledge will render the





resulting model uncertain, and quantifying such uncertainty is essential to making better use of the models, and also to making better predictions.

A common approach for building geological models is cognitive modeling (*e.g.* Jørgensen et al., 2013). Here, the dataset containing borehole lithology logs and geophysical models are co-interpreted by a professional with experience in the fields

of geoscience, geophysics, and geological modeling, with a relevant regional conceptual model in mind. This modeling approach is deterministic, and results in a single model realization, thus making it difficult to quantify the related structural uncertainty. Furthermore, using the geological model to make predictions, the uncertainty related to the geological model carries over into the predictions. If the forecasts are based on a single geologic model, the prediction does not encase the full complexity of the problem. Alternatively, if the model uncertainty can be quantified, it enables the option to include it in the

forecast. However, quantifying the uncertainty in such a cognitive modeling approach is difficult and tedious. Another approach is stochastic modeling using Multiple-Point Statistics (MPS) methodologies. MPS provides a framework which can integrate geophysical and borehole information, as well as conceptual geological information via a so-called Training Image (TI). Multiple model realizations are created from the dataset. The resulting model ensemble reflects the uncertainty related to the underlying datasets and overall modeling procedure.

Geophysical data is spatially dense and provides a smeared image of the overall subsurface geology. Resolution decreases with depth, and diminishes at a specific depth which is dependent on the geophysical method. Geophysical instruments portray bulk physical properties of the subsurface. Although geophysical data provides spatially dense information, it is not possible to exhaustively sample our model. The density of the geophysical data will affect the final uncertainty. The raw geophysical data goes through a processing and modeling step, where the raw data are translated into geophysical models.

During this step incorrect measurements, due to instrument error or interference, are identified and removed, further decreasing the geophysical information density. Presently, two approaches are taken when it comes to incomplete geophysical data. A common approach is to reconstruct the incomplete dataset using geostatistical tools (*e.g.* Goovaerts, 1997; Mariethoz and Renard, 2010). However, in this case it is important to emphasize that the reconstructed information is not as valuable as the actual measured geophysical information.

Borehole lithology logs are commonly viewed as "ground truth". However, lithology logs are also uncertain. In this study the boreholes are divided into 5 quality groups, of which only boreholes above a chosen threshold are used. Generally the uncertainty of borehole lithology logs relates to a number of parameters, such as: drilling methods, the frequency with which sediment samples are collected, precision with which the location is measured, the purpose of the borehole, the choice of drilling contractor *etc.* – see Barfod *et al.* (2016) and He *et al.* (2014) for more detail. The resolution of borehole lithology

logs is especially dependent on the sampling method. If a core is extracted for the entirety of the borehole, the resolution is, in principal, unlimited. However, this is expensive. It is more common to use either an auger drill, rotary drill or a cable tool, which yields a relatively limited resolution, compared to core drilling, depending on how samples are collected and handled.




In this study, the "Single Normal Equation Simulation" (*snesim*) MPS framework (Strebelle, 2002) is used to create geologic models and study the uncertainty related to the geophysical data, lithology logs, and conceptual geological model (TI). It is carried out on a hydrogeophysical dataset from Kasted, Denmark. Since subsurface hydraulic flow is largely controlled by geological heterogeneity (*e.g.* Feyen and Caers, 2006; Fleckenstein et al., 2006; Fogg et al., 1998; Freeze, 1975; Gelhar, 5 1984; LaBolle and Fogg, 2001; Zhao and Illman, 2017; Zheng and Gorelick, 2003), accurate geological models are crucial to accurate predictions of hydraulic flow. Geological units, however, contain additional complexities not related to hydrologic units; therefore, from here on, the concept of hydrostratigraphic units will be used. A detailed definition of hydrostratigraphic classification is given by Maxey (1964).

We present a study of the uncertainty related to stochastic hydrostratigraphic MPS modeling of a hydrogeophysical dataset 10 from Kasted, Denmark. The goal is to understand the consequences of modifying the underlying MPS setup to reflect some of the biases related to a real-world hydrogeophysical dataset and study the propagation of the uncertainty into the hydrostratigraphic models. We show how uncertainty related to resistivity data, measured with the SkyTEM system (Sørensen and Auken, 2004), and borehole lithology logs influences the hydrostratigraphic modeling realizations. Two readily available MPS tools are showcased. The first tool is the Direct Sampling (DS) method for reconstruction of 15 incomplete datasets (Mariethoz and Renard, 2010). The other MPS tool is *snesim*, which is used for stochastic hydrostratigraphic modeling. The stochastic models will be divided into 6 overall cases. The first case is the basic modeling setup, which uses SkyTEM resistivity models as soft data, boreholes as hard data and a cognitive 3D hydrostratigraphic model as a TI. The remaining cases are then modified versions of the basic modeling setup, which are designed to reflect different types of modeling uncertainty: Case 1 – Conceptual geological understanding, Case 2 – Incomplete soft data, Case 20 3 – Choice of resistivity model, Case 4 – Borehole lithology logs, and Case 5 – No soft data.

A total of 400 MPS realizations are created. Comparing such a large number of 3D hydrostratigraphic realizations visually is difficult to do in a quantitative manner. Therefore, a mathematical comparison method based on Euclidean Distance Transforms (EDT) (Maurer et al., 2003) is used. The EDT converts a 3D binary image to a continuous 3D Euclidean distance grid. Two 3D EDT grids are then compared by calculating the average difference in the respective grids. Similar 25 images have a small average EDT distance, and dissimilar images have a large average EDT distance. The EDT-based distance results are computed for all of the 400 realizations, and grouped by case, to create a full distance matrix, comparable to a co-variance matrix. An example is Case s4, which studies the effect of boreholes on the hydrostratigraphic realizations. Two MPS realization ensembles are created, 50 realizations with boreholes included, and 50 realizations without borehole information. The two model ensembles can be compared in two ways: visually and mathematically. Visual comparison of 30 the realization ensembles is tedious and subjective, but offers an overall understanding of the geological realism of the models (Barfod et al., 2017). The mathematical comparison provides a quantitative overview of similarity, or dissimilarity, of the two realization ensembles.



## 2 Materials and methods

### 2.1 The Kasted study area

The Kasted survey area is located north-west of Aarhus, Denmark (Figure 1A), and has also presented by Barfod *et al.*
(2017), Marker *et al.* (2017), and Høyer *et al.* (2015). The regional geology of the Kasted area is dominated by a Quaternary

buried valley complex with complex abutting relationships between the individual valleys. The buried valleys are infilled
with a combination of till and glacial meltwater deposits. The valleys are incised into the substratum which consists of
hemipelagic clay. The regional geology has been described in detail by Høyer *et al.* (2015), who created a detailed cognitive
geological model of the area.

The survey covers an area of 45 km$^2$, and is composed of a 333 line km spatially dense SkyTEM survey (Sørensen and

Auken, 2004), with a line spacing of 100 m. The resulting SkyTEM soundings have been processed according to the
description by Auken *et al.* (2009). Finally two sets of geophysical models were produced using either the smooth Spatially
Constrained Inversion (SCI) models (Constable et al., 1987; Viezzoli et al., 2008), or the sharp SCI (sSCI) models (Vignoli
et al., 2015). Furthermore, there are 948 boreholes scattered throughout the greater Kasted survey area, each with a
corresponding lithology log of a varying quality. Only 410 of the boreholes are above the selected quality threshold, and

contain lithological information relevant to this study. An overview of the dataset is found in Figure 1C and described in
further details in Barfod *et al.* (2017) and Høyer *et al.* (2015).



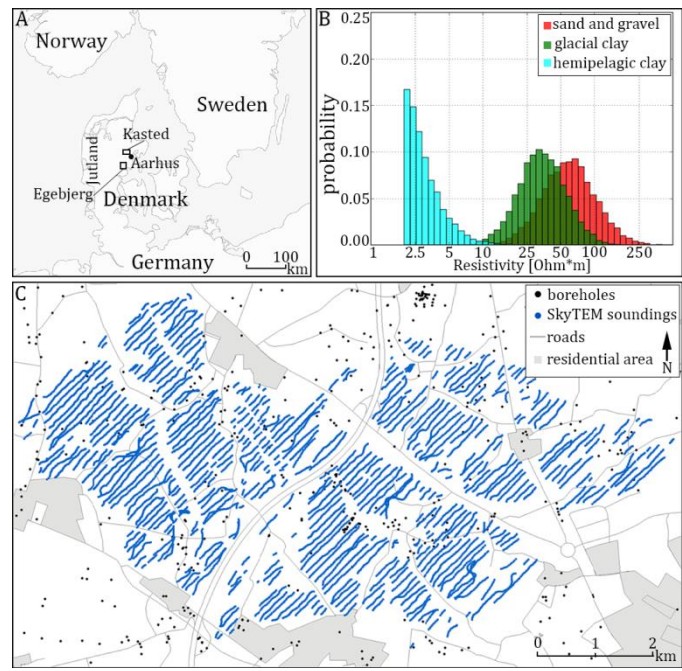

**Figure 1:** *The Kasted survey area and resistivity-hydrostratigraphic relationship histograms. **A** shows the geographical location of the Kasted survey area and the Egebjerg model used as a secondary TI. **B** shows the reconstructed resistivity-hydrostratigraphic relationship histograms for the three main hydrostratigraphic unit categories based on SkyTEM resistivity models and borehole lithology logs. **C** shows the Kasted survey with the SkyTEM sounding and borehole locations.*

## 2.2 Multiple-Point Statistics (MPS) and single normal equation simulation (*snesim*)

The Multiple-Point Statistics (MPS) framework stems from the, more general, geostatistics framework. Here, Multiple-Point (MP) information from a Training Image (TI) is used to condition simulations to probable patterns from the 2D or 3D digital TI (Journel and Zhang, 2007). The TI provides a conceptual geological understanding of a given area. The TI can be viewed as a database containing realistic geological patterns. This MP information is used to condition the simulation, to increase the overall geological realism of the realizations. The choice of TI is an important step in any MPS setup, and influences the realization results, as will be illustrated. The TI does not need to carry any locally accurate information, *i.e.* the TI does not need to spatially or geographically overlap with real-world geological units, and can be purely conceptual in nature. Together with the TI, it is also possible to use geophysical datasets for constraining MPS simulations, resulting in realizations reflect real-world regional geology. Today, MPS is a widely used tool, which is used in a variety of geoscience fields, including, but not limited to: reservoir modeling (*e.g.* Okabe and Blunt, 2004; Strebelle and Journel, 2001), hydrology (*e.g.* Le Coz et al., 2011; Hermans et al., 2015), geological modeling (e.g. Høyer et al., 2017; de Iaco and Maggio, 2011) *etc*.

The MPS method used in this paper is known as the single normal equation simulation (*snesim*) framework (Strebelle, 2002), and is implemented in the Stanford Geostatistical Modeling Software, or SGeMS. The *snesim* framework allows for simulating real-world categorical geological model using a TI, constrained using soft geophysical data, and hard borehole





data. The *snesim* algorithm scans the entire *TI*, ahead of simulation, and stores the MP information contained in the TI in a search-tree database. The *MP* information can then be retrieved from the database during simulation. The integration of soft geophysical data for constraining the simulations is achieved by utilizing the tau model (Journel, 2002; Krishnan, 2004). Here, the continuous soft data variable needs to be translated into a probability grid, describing the probability of finding

given geological unit based on the geophysical data; see Barfod *et al.* (Barfod et al., 2017). In order to guarantee the reproduction of geological patterns at all scales, *snesim* uses the multiple grid formulation, presented by Tran (1994).

### 2.3 *MPS* modeling setup

The Kasted dataset is comprised of a dense geophysical SkyTEM dataset (Sørensen and Auken, 2004), borehole lithology logs and a cognitive geological model (Høyer et al., 2015). The MPS modeling setup is similar to the one presented by

Barfod *et al.* (Barfod et al., 2017). However, the goal of this study is different. Since a 3D deterministic model of the area already exists, the study area, and overall geological setting, is well known. Additionally, the cognitive geological model can be utilized as a 3D TI. The Kasted model covers an area of 12 km by 7 km, discretized on a modeling grid with 229 by 133 by 39 cells, containing a total of 1,187,823 cells. Each cell has a size of 50 m by 50 m by 5 m. The Kasted survey lithology logs reveal a combination of 59 geological categories, which are grouped together into three key hydrostratigraphic units:

1) *sand and gravel*: a combination of coarse lithological units, including sand till, meltwater sand, gravel and pebbles of glacial origin, late glacial  freshwater sand and postglacial freshwater sand

   2) *glacial clay*: this category contains silty and sandy clays, including clay till and meltwater clay of glacial origin

   3) *hemipelagic clay*: a combination of fine grained conductive clays, containing the extensive and homogeneous hemipelagic Paleogene and Oligocene clays found in Denmark

These three categories serve the purpose of simplifying the geology of the Kasted area. Using these hydrostratigraphic categories, the lithology logs are translated into a set of hydrostratigraphic logs ("Step1", Figure 2), and the cognitive geological model is translated into a hydrostratigraphic model (Figure 3A).

The 1D SkyTEM resistivity models are assigned to a 3D grid, identical to the modeling grid, by using block Kriging, which is different from the approach in Barfod *et al.* (2017). Originally, in Barfod *et al.* (2017) a simple Kriging estimation

approach was used to assign the resistivity models to a 3D modeling grid. This resulted in an incomplete resistivity grid which contained resistivity information not only pertaining to grid cells containing a SkyTEM sounding. This meant that the resistivity grid had already been slightly reconstructed. To avoid this, block Kriging estimation is used instead. The block Kriging method is also a variogram based estimation method, which estimates the average value of a rectangular block (Goovaerts, 1997). The block average resistivity value for all grid cells containing a resistivity model is estimated. The end

result is an incomplete resistivity grid (Figure 5A). The incomplete resistivity grid is then reconstructed using DS stochastic





reconstruction (Mariethoz and Renard, 2010) (Figure 6). See Barfod *et al.* (2017) for more detail on reconstruction of the incomplete resistivity grids of the Kasted survey.

The SGeMS *snesim* framework utilizes the tau model for soft data conditioning (Journel, 2002), which requires the translation of resistivity grids into probability grids. This requires information on the regional resistivity-hydrostratigraphic

relationship. Such knowledge is not always available, but if enough boreholes and electromagnetic geophysical data is available, the framework for studying the resistivity-hydrostratigraphic relationship, presented by Barfod *et al.* (2016), can be used to create a set of histograms. The resistivity-hydrostratigraphic histograms, Figure 1B, are compiled from available hydrostratigraphic logs and SkyTEM resistivity models, and are presented in more detail in Barfod *et al.* (2016). The estimated histograms (Figure 1B) are then used to directly translate each resistivity value, in a given resistivity grid, into

three probabilities, one for each hydrostratigraphic unit.

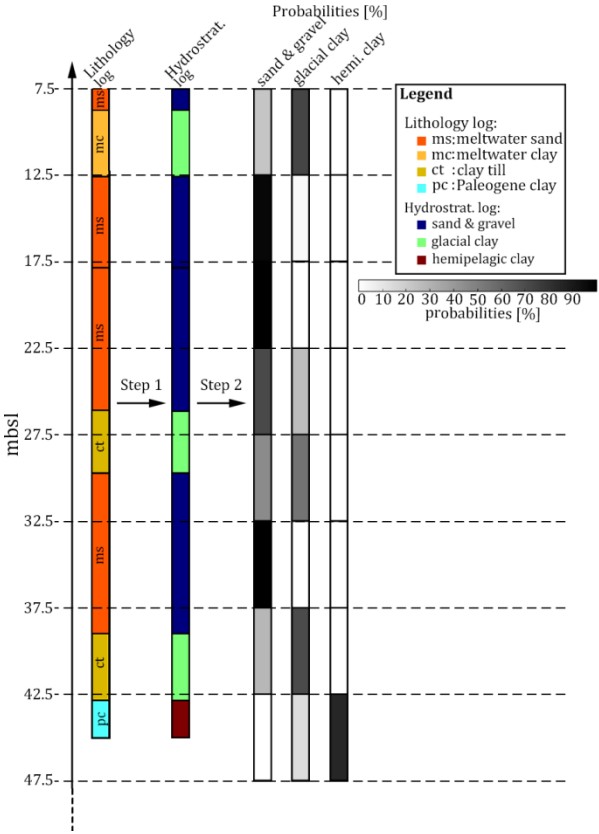

**Figure 2:** *A schematic diagram presenting the conversion of the lithological logs into probability logs for the three hydrostratigraphic units: sand & gravel, glacial clay and hemipelagic clay. Step1: the lithology log is translated into a hydrostratigraphic log. Step 2: The hydrostratigraphic logs are resampled according to the vertical modeling grid intervals and an interval probability is calculated for each*

*of the hydrostratigraphic units.*





Like the resistivity models, the borehole lithology logs also need to be assigned to a 3D grid, which is carried out in three overall steps. The first step is to translate the borehole lithology logs into hydrostratigraphic logs using prior knowledge regarding the regional hydrostratigraphy; "Step 1" Figure 2. The second step is then to divide the hydrostratigraphic logs into intervals identical to the vertical intervals of the model grid. At each resampled interval a probability value is directly

calculated for each hydrostratigraphic unit; "Step 2" in Figure 2. Finally, the last step is to assign the hydrostratigraphic probabilities to a grid. The probability values are assigned to the grid cell in which the given hydrostratigraphic log is present. On the rare occasion that multiple logs are present within a given cell, the probabilities are combined accordingly to one representative probability value. The end result is a grid containing the borehole probability values of each hydrostratigraphic unit: *sand & gravel*, *glacial clay* and *hemipelagic clay*. It is common to view borehole lithology logs as

hard information, or "ground truth" (e.g. Gunnink and Siemon, 2015; Tahmasebi et al., 2012). The borehole probability grid can therefore be translated into a hard data grid, by assigning the most probable hydrostratigraphic unit in each grid cell.

The general MPS setup can be summarized in 7 overall steps as follows:

1) Using block Kriging, the SkyTEM resistivity models are assigned to a 3D grid identical to the Kasted model grid.

2) The incomplete resistivity grids are stochastically reconstructed using Direct Sampling (DS), as presented by

Mariethoz and Renard (2010) (Figure 5A and Figure 6A). The result is an ensemble of 50 equiprobable reconstructed resistivity grids.

3) The reconstructed resistivity grids are translated into probability grids using the resistivity-hydrostratigraphic relationship histograms (Figure 1B).

4) The borehole lithology logs are translated into hydrostratigraphic logs; "Step 1" Figure 2.

5) The hydrostratigraphic logs are resampled and three probability values, one for each hydrostratigraphic unit, is directly computed at each resampled interval; "Step 2" Figure 2.

6) The borehole probabilities are assigned to a grid identical to the cognitive Kasted model grid.

7) The borehole probability grid is translated into a hard data grid, by assigning the most likely hydrostratigraphic unit to each grid cell.

This study is divided into a total of 6 cases, or 8 sub-cases, which are designed to study how perturbations of the underlying MPS setup affects the hydrostratigraphic realizations using *snesim*. A total of 400 realizations are created, with 50 realizations per sub-case – see Table 1 for more details and a brief description of each case. In *snesim* a random number seed needs to be manually selected for each realization to define a random path through the modeling grid. The random seed convention chosen in this paper was to apply the same random seed vector to each sub-case. The vector contains 50 linearly

increasing random seed numbers, ensuring consistency when comparing realizations from the individual sub-cases.



**Table 1.** *An overview table showing information on the MPS cases along with information on number of realizations for each case / sub-case, and a brief description of each case.*

| Case name | Sub-case names | Num. realizations | Total num. realizations | Case description |
|---|---|---|---|---|
| Basic setup | Basic modeling setup | 50 | 50 | *The basic setup uses boreholes as hard data, smooth resistivity models as soft data, and the cognitive Kasted model as a TI* |
| Case 1a | a) Egebjerg TI | 50 | 100 | *Two different TIs are used to study the uncertainty related to the choice of TI, which reflects the conceptual geological understanding* |
| Case1b | b) Conceptual TI | 50 | | |
| Case 2 | Incomplete soft data grid | 50 | 50 | *The uncertainty related to the reconstruction of the resistivity grid is studied by running simulations with an incomplete resistivity grid* |
| Case 3 | Sharp resistivity models | 50 | 50 | *The sharp resistivity models are used for simulations instead of the smooth models, to study how the choice of resistivity model influences the hydrostratigraphic models* |
| Case 4a | a) No borehole data | 50 | 100 | *Simulations are run without hard data, to see how much the hard data influences the results* |
| Case 4b | b) Soft borehole data | 50 | | *The borehole data are used as soft information instead of hard by combining the borehole probability grid with the SkyTEM probability grid using the Tau model* |
| Case 5 | No soft resistivity data | 50 | 50 | *Simulations are run using only the hard data and the cognitive Kasted TI* |
| Total | --- | --- | 400 | --- |

### 2.3.1 Basic modeling setup

The basic modeling setup is designed to act as the base from which all other cases are built. The different sub-cases are
simply modified versions of the basic modeling setup, each designed to study how modification to the base setup relates to hydrostratigraphic MPS modeling. The basic modeling setup uses the borehole data as hard information, SCI models with smooth inversion constraints, and the cognitive hydrostratigraphic Kasted model as a TI (Figure 3A) (The Egebjerg model consists of a total of 72 geological units which are categorized accordingly to reflect the three hydrostratigraphic units of the Kasted hydrostratigraphic model. Egebjerg additionally contains undesired features, such as local Miocene complexes. Two
such local geological environments, which do not reflect the geological setting of the Kasted area, are present. One is found south of the buried valley complex, and the other to the west. By cropping the model and rotating it 90 degrees counter-clockwise, a relevant TI without undesired geological architecture is produced (Figure 3C), this is referred to as Case 1a. It is clearly seen, by comparing Figure 3A and C that the Kasted and Egebjerg TIs are different. The Kasted TI is smaller, and contains smooth geological features, while the Egebjerg model is larger and contains coarse, block-like geological features.
The important features, in relation to hydrostratigraphic modeling, are the buried valley complexes, which are present in the Egebjerg model (Figure 3C). The global proportions of the Egebjerg TI (Table 2) are similar to the ones found in the Kasted




TI. However, the vertical proportions of the Egebjerg TI (Figure 4C) are different, especially in the upper part of the TI where *glacial clay* units dominate.

Table 2) (Figure 4A). The hard borehole logs are created from hydrostratigraphic probability logs by assigning the most probable hydrostratigraphic unit to a hard data grid.

## 2.3.2 Case 1 – Conceptual geological understanding

The basic modeling setup uses the actual cognitive geological model of the Kasted survey area as a TI (Høyer et al., 2015). In Denmark, it is common practice to build 3D cognitive geological models of the near-subsurface. Many cognitive models exist and are publicly available. Such models can easily be adapted and used as 3D TIs to simulate new survey areas, provided the geological settings are similar. Case 1 is divided into two sub-cases. The first sub-case, Case1a, uses the basic setup, but in place of the cognitive Kasted model, the cognitive geological model of the Egebjerg area (Figure 1A) is used as a TI (Figure 3C). The geologic setting in Egebjerg is relevant since it is partly dominated by a buried valley complex (Jørgensen et al., 2010).



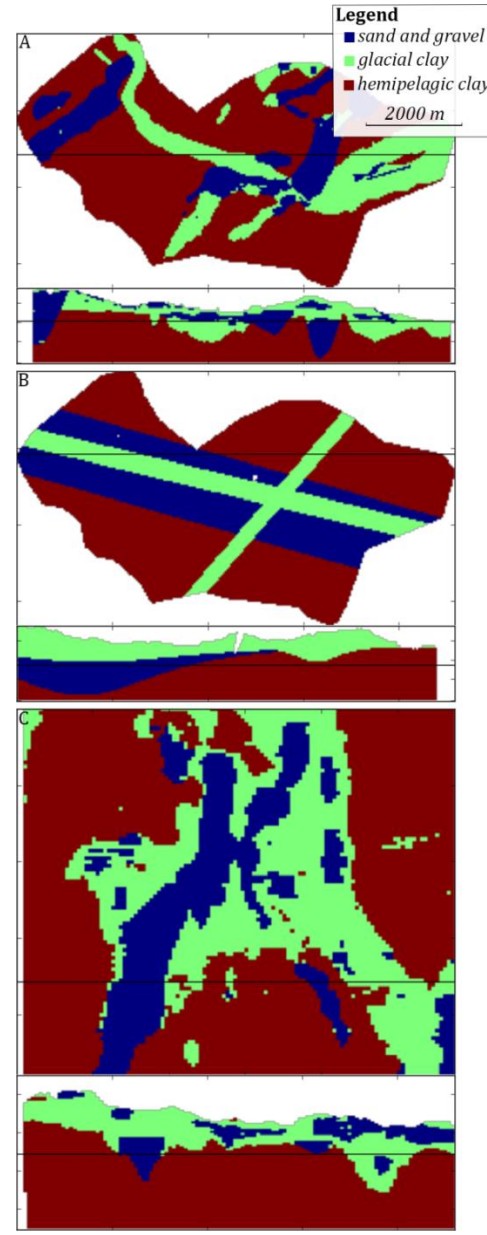

**Figure 3:** *An overview of the training images (TIs) which are used during MPS simulation. A horizontal cross-section and vertical slice is presented for each TI, portraying the hydrostratigraphic architecture; **A** shows the Kasted TI, **B** shows the conceptual TI, and **C** shows the Egebjerg TI.*

5     The Egebjerg model consists of a total of 72 geological units which are categorized accordingly to reflect the three hydrostratigraphic units of the Kasted hydrostratigraphic model. Egebjerg additionally contains undesired features, such as local Miocene complexes. Two such local geological environments, which do not reflect the geological setting of the Kasted area, are present. One is found south of the buried valley complex, and the other to the west. By cropping the model and





rotating it 90 degrees counter-clockwise, a relevant TI without undesired geological architecture is produced (Figure 3C), this is referred to as Case 1a. It is clearly seen, by comparing Figure 3A and C that the Kasted and Egebjerg TIs are different. The Kasted TI is smaller, and contains smooth geological features, while the Egebjerg model is larger and contains coarse, block-like geological features. The important features, in relation to hydrostratigraphic modeling, are the buried valley

complexes, which are present in the Egebjerg model (Figure 3C). The global proportions of the Egebjerg TI (Table 2) are similar to the ones found in the Kasted TI. However, the vertical proportions of the Egebjerg TI (Figure 4C) are different, especially in the upper part of the TI where *glacial clay* units dominate.

**Table 2:** *The global proportions related to each of the three TIs presented in Figure 3.*

|  | *sand and gravel* | *glacial clay* | *hemipelagic clay* |
|---|---|---|---|
| Kasted *TI* | 0.17 | 0.21 | 0.62 |
| Conceptual *TI* | 0.17 | 0.22 | 0.61 |
| Egebjerg *TI* | 0.10 | 0.22 | 0.68 |

The second sub-case, Case1b, utilizes a purely conceptual TI. The conceptual TI is created by using a set of hyperbolic

secant functions to populate a 3D matrix and is purely mathematical in nature. The conceptual TI can be seen in Figure 3B and is designed to have three overall buried valleys eroded into a *hemipelagic clay* substratum. There are two narrow and shallow *glacial clay* valleys, and a broad and deep *sand and gravel* valley. One of the glacial *clay valleys* is a younger valley which is eroded into the older *sand and gravel* valley, and run roughly parallel to each other. The last *glacial clay* valley is almost orthogonal to the other valleys, and also erodes into the *sand and gravel* valley. The upper part of the TI contains a

cover layer of glacial clay (Figure 4B). The simple conceptual TI is designed to contain the main geological architecture of the Kasted area, namely the buried valley complexes. The *sand and gravel* valley, trending west northwest – east southeast, was chosen on purpose to study what happens when over-simplified and smooth MP information is added to a TI. The global proportions of the conceptual TI are consistent with the other TIs, while the vertical proportions for *sand and gravel* and *glacial clay* units show a significantly different pattern (Figure 4B).

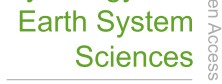



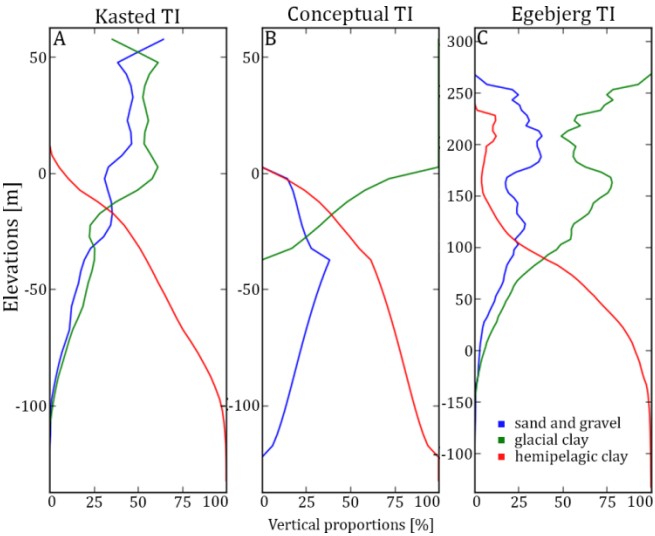

**Figure 4.** The vertical proportions of the three training images for each of the hydrostratigraphic categories, where **A** portrays the Kasted TI, **B** the conceptual TI, and **C** the Egebjerg TI.

### 2.3.3 Case 2 – Incomplete soft data

During reconstruction of the resistivity grid, it is assumed that the patterns in the incomplete data set contain information regarding the content of the data set gaps. This is true only when the incomplete grid contains a sufficient amount of data. Sufficient, in this case, means that the parameter space is sampled densely enough to reflect the patterns we wish to reconstruct (Mariethoz and Renard, 2010). If the grid is too sparse, then limited or no information is present which can help reconstruct missing patterns is present. Signs of mediocre data density are seen in the incomplete grids (Figure 5A). Artifacts

from the DS reconstruction are present in the completed resistivity grids. The resistive valley to the west in the horizontal slices and vertical cross-sections in Figure 6A and B reveals a striated pattern. An alternative to reconstructing the resistivity grid beforehand is to use the incomplete resistivity grids for simulation, meaning no information is present in the resistivity dataset gaps. Grid cells containing a resistivity model are translated into three probability values using the resistivity-hydrostratigraphic relationship histograms (Figure 1B) (Figure 5B-D). Areas without soft resistivity data rely on the TI

during simulation, emphasizing the fact that no actual information is present between soundings. The overall setup is identical to the basic setup; the only difference is the reconstructed soft data grids are interchanged for the incomplete soft data grid (Figure 5B-D).





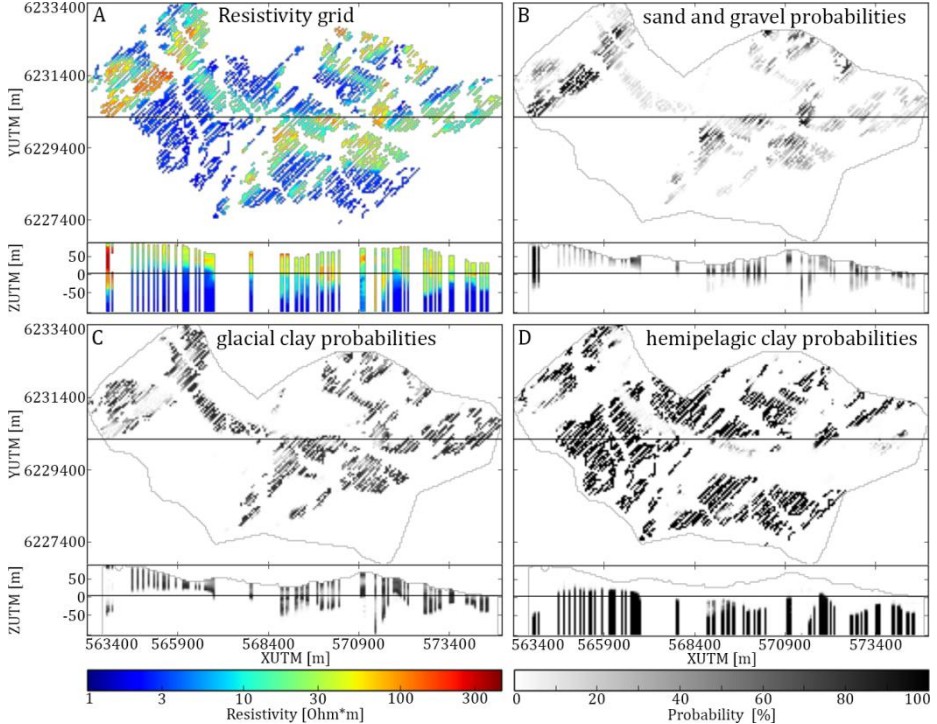

**Figure 5:** *A presentation of the incomplete resistivity grid. Each grid is portrayed as a horizontal cross-section at 20 mbsl, and a vertical slice intersecting at UTMY 6230100 m. **A** shows the resistivity grid which is translated into three probability grids using the resistivity-hydrostratigraphic relationship histograms (Figure 1B). Grid cells without SkyTEM soundings are not assigned a probability value. **B-D** show the sand and gravel, glacial clay, and hemipelagic clay probability values, respectively.*

### 2.3.4 Case 3 – Choice of resistivity model

The choice of inversion algorithm results in different SkyTEM resistivity models. The purpose of this case is to study how using sSCI (Vignoli et al., 2015) models influences the modeling results. A common inversion approach is SCI where a smooth regularization is used (Constable et al., 1987). Such resistivity models have a smooth transition from resistive to conductive features, and *vice versa*. Geological layer boundaries are rarely smooth in nature, meaning such soft transitions in resistivities seldom reflect reality. Furthermore, extreme resistivity values are not presented correctly in the smooth model inversions. Vignoli *et al.* (2015) propose an alternative SCI approach, employing a "Minimum Gradient Support" regularization term instead. Such sSCI models produces resistivity models with sharp layer boundaries and a better representation of extreme values. The setup in Case 3 is identical to the basic setup, except that the SCI models are interchanged for sSCI models. The DS grid reconstruction is then conducted on the sSCI models, which are then translated into probability grids. Finally, these grids are used as soft data for simulation using the *snesim* method.

The sharp resistivity models are different from the smooth models, but no particularly sharp layer boundaries are reflected in the reconstructed resistivity grid (Figure 6B).





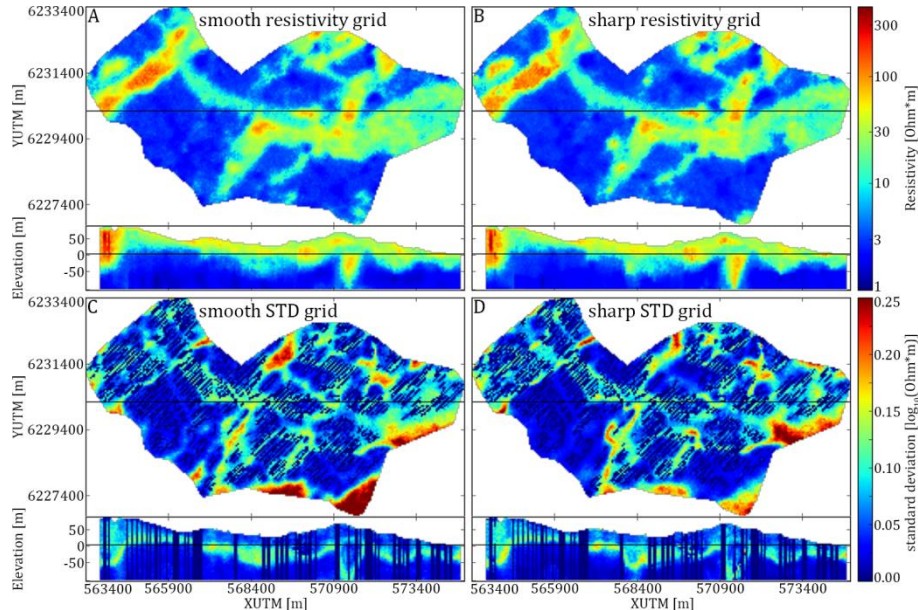

**Figure 6:** *An overview of the key differences between reconstructing the resistivity grid using smooth and sharp inversion resistivity models. Each grid is portrayed as a horizontal cross-section at 20 mbsl, and a vertical slice intersecting at UTMY 6230100 m. **A** shows the smooth reconstructed resistivity grid. **B** portrays the sharp reconstructed resistivity grid. **C** shows the standard deviation calculated from 50 stochastic reconstructions of the smooth resistivity grid. **D** shows the standard deviation calculated from 50 stochastic reconstructions of the sharp resistivity grid.*

One of the obvious differences is found in the resistivity patterns of the *sand & gravel* valley to the far west of the survey area. The valley itself is not significantly different, however, the small resistive patch, west of the large valley, is more pronounced in the sharp model and has an overall more pronounced fingerprint (Figure 6B). The sharp resistivity models better estimate the "true" bulk resistivity values of specific geological units, such as the resistive patch accentuated here. The ensemble standard deviation grid, Figure 6C and D, show a general reduction in the ambiguity of the reconstructed sharp resistivity models. This is clear from the reduction areas with large standard deviation, red colors, which are overall reduced in size.

### 2.3.5 Case 4 – Borehole lithology logs

This case is dedicated towards how the borehole data is handled, and how it influences the hydrostratigraphic modeling results. The hard borehole data is normally sparse, relative to geophysical data. Boreholes are commonly considered "ground truth" since they directly sample the subsurface sediments or petrological units. This case is divided into two sub-cases. The first sub-case, Case 4a, portrays what happens when hard data is not included in the *snesim* simulation. The model setup is therefore identical to the basic MPS setup, but without including the borehole data.

The second sub-case, Case 4b, incorporates the borehole lithology logs as soft data. The certainty of a lithological log varies depending on a range of factors, *e.g.* drilling method, the purpose of the borehole, and sampling frequency (*e.g.* Barfod et al.,



2016; He et al., 2014). The hydrostratigraphic probability logs, introduced in the basic modeling setup ("Step 2" Figure 2), are utilized in place of the hard borehole grid. The boreholes are assigned a lateral footprint, so the information is not only found at the borehole locations. The borehole footprint is assigned by creating a grid where the borehole probability values have been estimated in a radius of 200 m around each borehole using simple Kriging with a search radius of 200 m and a

mean of 1/K=1/3, where K is the number of unique hydrostratigraphic units (Figure 7D-F). The tau model is then used to combine the SkyTEM (Figure 7A-C) and borehole (Figure 7D-F) probability grids (e.g. Journel, 2002; Krishnan, 2004; Remy et al., 2014). The usage of the Tau model for combining soft data grids will be briefly described here. Suppose we have a set of data events, $\boldsymbol{D}_i, i = 1, \dots, n$, and the goal is to estimate the probability that a hydrostratigraphic unit (A) is present provided all data events:

$P(A|\boldsymbol{D}_1, \dots, \boldsymbol{D}_n)$         (1)

The first step is then to define the prior probability distribution, $P(A)$, which in this case are the vertical proportions taken from each layer of the cognitive Kasted TI (Figure 4A). Then the probability distributions, which are to be combined, are defined: $P(A|\boldsymbol{D}_1)$ and $P(A|\boldsymbol{D}_2)$, where $\boldsymbol{D}_1$ is the resistivity probability grid and $\boldsymbol{D}_2$ the borehole probability grid. The 3D probability grids are translated into distance grids by applying the "*probability-into-distance*" transform:

$x_o = \frac{1-P(A)}{P(A)}, x_1 = \frac{1-P(A|\boldsymbol{D}_1)}{P(A|\boldsymbol{D}_1)}$, and $x_2 = \frac{1-(A|\boldsymbol{D}_2)}{P(A|\boldsymbol{D}_2)}$

Then the following distance ratio is computed using the tau model expression:

$\frac{x}{x_0} = \prod_{i=1}^{n=2} \left(\frac{x_i}{x_0}\right)^{\tau_i}, \tau \epsilon [-\infty; +\infty]$         (2)

where the tau values are chosen as follows: $[\tau_{resistivity}, \tau_{borehole}] = [2,1]$. The final conditional probability is computed as follows:

$P(A|\boldsymbol{D}_1, \boldsymbol{D}_2) = \frac{1}{1+x}$         (3)

where the value of x is computed from Eq. (2), as follows:

$x = x_0 \cdot \left(\frac{x_1}{x_0}\right)^{\tau_{resistivity}} \cdot \left(\frac{x_2}{x_0}\right)^{\tau_{borehole}}$

The resulting combination of the three hydrostratigraphic probability grids is seen in Figure 7G-I. The combined probability grids are used in the basic setup, in place of the smooth probability grids.

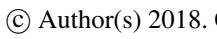



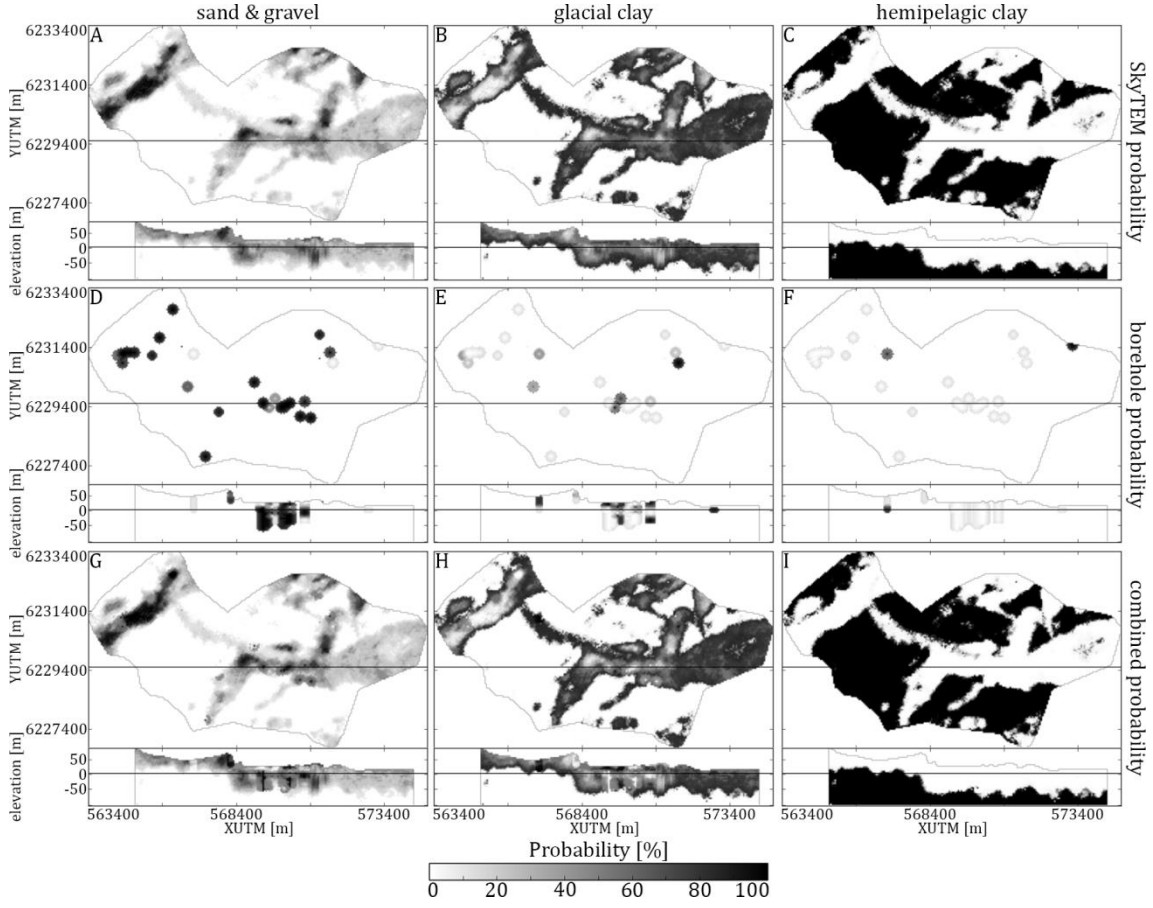

**Figure 7:** *A visual representation of the Tau model procedure for combining the soft resistivity and borehole grids. Each grid is portrayed as a horizontal cross-section at 20 mbsl, and a vertical slice intersecting at UTMY 6230100 m; **A-C** shows the sand and gravel, glacial clay, and hemipelagic clay probability maps, respectively, for one DS reconstructed resistivity grid, **D-F** shows the 200 m radius Kriged borehole probability, and **G-I** shows the combined resistivity grid which has been combined using a tau model with the values:[$\tau_{resistivity}$, $\tau_{borehole}$ ]=[2,1].*

### 2.3.6 Case 5 – Excluding the soft resistivity data

The final case, Case 5, illustrates the consequences of not including the soft SkyTEM resistivity information in the MPS simulation routine. The basic setup is simply run without the inclusion of soft data, *i.e.* the setup only uses the cognitive Kasted TI and hard borehole information.

### 2.4. Comparing Simulation results

Comparing a large set of extensive 3D models is a common problem encountered in stochastic MPS modeling. A common approach is visual comparison, which is not an objective or quantitative comparison method. Each equiprobable hydrostratigraphic model in this study contains 1,187,823 cells. Furthermore, a total of 400 MPS realizations were





computed, Table 1, which makes it difficult to visually compare modeling results. This, along with advances in stochastic modeling tools such as MPS, motivated Tan *et al.* (2014) to develop a framework in which multiple 2D or 3D realizations can be compared  quantitatively. The idea is to use a distance measure, which measures the distance between two realizations. Realizations which are geometrically similar have small distance values, while dissimilar realizations have a large distance value. The comparison techniques in this study are based on the principles presented by Tan *et al.* (2014). In this study the distances between individual realizations are based on the Euclidean Distance Transforms (EDT) (Maurer et al., 2003). The usage of EDT as a measure for similarity will be described in more detail below. A full distance matrix is computed containing distances between each individual realization for all the different cases. The resulting 400 by 400 distance matrix is then interpreted by itself.

## 2.4.1 Ensemble mode ratio maps (EMR-maps)

The visual comparison can be helped by creating so-called Ensemble mode ratio maps, or EMR-maps. The idea is to create a summary map portraying the mode ratio of a given ensemble of models, ranging between 1/K and 1, where K is the number of hydrostratigraphic categories. The EMR-maps describe the certainty of the simulation based on the resulting realization ensemble. If the EMR-map shows a value of one, then every single realization in the present ensemble has simulated the same category or, in this case, hydrostratigraphic unit. On the other hand if the EMR-map shows a ratio of 1/K the ensemble of realizations shows equal probability for each of the K categories. Each realization is equiprobable, and the EMR values of the categorical variables are computed from the probability distribution of a given cell with location, $\boldsymbol{u}$. The probability that the attribute S is equal to $s_k$, $P_k(\mathbf{u})$, which is computed as follows:

$$P_k(\boldsymbol{u}) = \frac{1}{N_{reals.}} \sum_{i=1}^{N_{reals}} (s_{k,i}(\boldsymbol{u}) == s_k) \tag{4}$$

where $s_k$ is the state of attribute S for which we are currently computing the probability and $s_{k,i}(\mathbf{u})$ is the state of the attribute at location $\mathbf{u}$ and for the $i$'th realization. The EMR values for a given cell, $\mathbf{u}$, can then be computed as follows:

$$r_{EMR}(\boldsymbol{u}) = \max_{k=\{1,2,...,K\}} (P_k(\boldsymbol{u})) \tag{5}$$

where K is the number of categories for which the EMR value is computed, and $P_k(\mathbf{u})$ denotes the probability for category $k$ at location $\mathbf{u}$ computed using eq. (4).

The EMR values are then computed for each grid cell using eq. (3) and (5), which, simply put, is the occurrence ratio of the mode category of a given ensemble containing a given number of realizations, $N_{reals}$. In other words, if at a given location, $\mathbf{u}$, if 45 out of 50 realizations yield the same category, then the EMR-value is 0.9, and the ensemble certainty for the given cell is high. On the other hand, with three possible lithological categories *i.e.* K=3, the lowest possible certainty is 1/K=1/3, which means there is an equal probability of occurrence for each lithological category. This means that $P(s_1)=P(s_2)=P(s_3)=1/3$, and therefore at the given location, $\mathbf{u}$, the $r_{EMR}=1/3$ and the simulation is uncertain.



### 2.4.2 Euclidean Distance Transforms (EDT) – measuring similarity between 3D hydrostratigraphic realizations

The hydrostratigraphic realizations are categorical and contain three hydrostratigraphic units. Comparing two realization grids, they first need to be transformed from a categorical grid into continuous Euclidean distance grids by using a EDT (Maurer et al., 2003). The EDT computes the Euclidean Distances for all locations of a binary grid, *i.e.* a grid containing only two states. For a given grid cell, with n-dimensional location vector $\boldsymbol{u}$, the Euclidean Distance is computed between location $\boldsymbol{u}$, and all other locations in the grid, $\boldsymbol{v}_i, i = \{1, \dots, N_{cells}\}$, with $\boldsymbol{u} \neq \boldsymbol{v}_i$ and $N_{cells}$ being the number of cells contained in the grid:

$$d_{EDT}(\boldsymbol{u}) = \min_{v \in V}(\|\boldsymbol{u} - \boldsymbol{v}\|_2) \tag{6}$$

The $d_{EDT}$ implementation presented by Maurer *et al.* (2003), uses a computationally favorable method for computing the exhaustive EDT at all locations in a binary grid.

To illustrate the $d_{EDT}$ approach for comparing realizations a 2D example case is presented. The basic modeling setup contains 50 realizations, *i.e.* $N_{realizations} = 50$, which are going to be compared to the cognitive model, which in this case also happens to be the TI. The 2D example is created by selecting the horizontal cross-section at 20 mbsl, for each of the 50 basic modeling setup realizations and the single cognitive geological model (Figure 8A-D). Each of the 2D layers are transformed into 2D binary layers, portraying *sand and gravel* as the main variable, and *glacial clay* and *hemipelagic clay* as a background variable (Figure 8E-H). The 2D binary layers are then translated into 2D $d_{EDT}$-layers by using eq. (6) to exhaustively compute the $d_{EDT}$ at each grid cell for all of the 50 realizations. The resulting $d_{EDT}$ layers, of which three are seen in Figure 8I-L, are used to compute an average Euclidean Distance between each realization, $realization_i$, and the cognitive geological model, $cog. model$:

$$\Delta \overline{d_{EDT}}(cog. model, realization_i) = \frac{1}{N_{cells}} \sum_{j=1}^{N_{cells}} \left[ d_{EDT}^{cog.model}(\boldsymbol{u}_j) - d_{EDT}^{realization_i}(\boldsymbol{u}_j) \right] \tag{7}$$

where $i \in \{1, \dots, N_{realizations}\}$, with $N_{realizations}=50$, and $N_{cells}$ being the number of cells in the, in this case, 2D layer. The $\overline{d_{EDT}}$, eq. (7), then describes the average difference of the distance to the nearest active cell in the binary grid. The 50 realizations are then ranked by the average Euclidean Distance differences, $\overline{d_{EDT}}$, as seen in the Figure 8, where the realization which is closest to the cognitive geological model (Figure 8B, F and J) has a $\overline{d_{EDT}}$-value of 240 m, while the realization which was ranked 25th closest (Figure 8C, G and K) has an $\overline{d_{EDT}}$-value of 280 m, and lastly the realization which was farthest (Figure 8D, H and L) has a $\overline{d_{EDT}}$-value of 310 m. It should be noted that the $\overline{d_{EDT}}$ computation, described by eq. (7), is not limited to comparing a realizations to a cognitive model, and can in fact be used to compare any pair of 3D categorical model. In fact a generalized version of eq. (7) can be defined as follows:

$$\Delta \overline{d_{EDT}}(model_A, model_B) = \frac{1}{N_{cells}} \sum_{j=1}^{N_{cells}} \left[ d_{EDT}^{model_A}(\boldsymbol{u}_j) - d_{EDT}^{model_B}(\boldsymbol{u}_j) \right] \tag{8}$$





Where the number of cells in model$_A$ must be equal to the number of cells in model$_B$, *i.e.* $N_{cells}^{model_A} = N_{cells}^{model_B} = N_{cells}$.

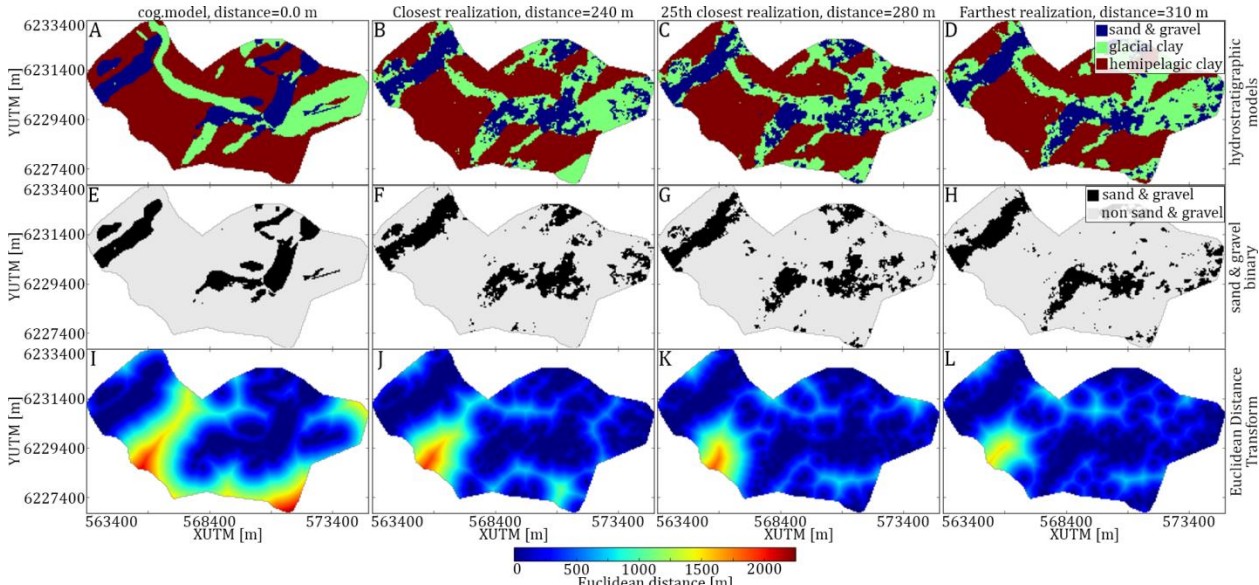

**Figure 8:** *A 2D example of the Euclidean Distance Transforms (EDT) as a measure for the similarity between categorical MPS realizations. In this example a set of 50 realizations, from the basic modeling setup, are compared based on the differences in EDT for sand & gravel units. **A-D** shows the hydrostratigraphic models for the TI, closest realization, 25th closest realizations, and farthest realization, respectively. **E-H** shows, in the same order as above, the binary images of the sand & gravel units of the 2D hydrostratigraphic model layers. **I-L** shows, in the same order as above, the Euclidean Distances layers computed from the 2D sand & gravel binary layers.*

From this point forward we leave the 2D example behind, and will from here on only consider $\overline{d_{EDT}}$ computations on 3D hydrostratigraphic grids. Furthermore, the $\overline{d_{EDT}}$ computations are carried out on a set of three binary grids, one for each of the three hydrostratigraphic categories. The distance value between two hydrostratigraphic grids is the summed distance for each of the three hydrostratigraphic categories, ensuring that the distance values reflect the complexities related to each of the hydrostratigraphic categories.

### 2.4.3 Evaluating the distance matrix

The average Euclidean Distance difference, $\overline{d_{EDT}}$, from here on referred to as the "distance" between two realizations, is exhaustively computed between all realizations and compiled into an exhaustive 400 by 400 distance matrix. The distance matrix, **D**, contains all distance values between all hydrostratigraphic realizations using Equation (6) and is defined as follows:

$$\boldsymbol{D}_{i,j} = \overline{d_{EDT}}\big(realization_i, realization_j\big)$$





where $i, j = \{1, \dots, N_{realizations}\}$. The distance matrix, **D**, can be evaluated directly by comparing the distances between individual realizations to each other. Another option is to summarize the distance matrix in a table representing the distances between the different cases. This is achieved by organizing the distance matrix according to which case they belong to. In this study the distance matrix is sorted according to the order of the individual cases, as in Table 1. The distance matrix can

then be summarized, by computing the average distance for each group of realizations pertaining to a specific case. The concepts of distance variability and distance to cognitive model were presented by Barfod *et al.* (2017), and are also used here. The concept is that the variability pertaining to a specific case can be computed by computing the average of the distances of the 50 realizations for a given case ensemble. Another measure is the distance to the cognitive model. The distances between all realizations and the cognitive model are computed, and provides a reference point to which the

realizations are compared.

## 3 Results

### 3.1 Visual comparison of hydrostratigraphic realizations and "Ensemble Mode Ratio"-maps (EMR-maps)

For each of the presented cases two hydrostratigraphic realizations are presented (Figure 9), along with an EMR-map (Figure 10). The EMR-maps show the occurrence ratio of the most likely simulated category for each grid cell based on 50

realizations. The two realizations and EMR-map of the basic modeling setup, Figure 9A and Figure 10A, reveal the same overall trends as the cognitive geological model, Figure 3A. Namely the western *sand and gravel* valley striking ~N40°E, the *glacial clay* valley striking ~E30°S, the large mixed *sand and gravel* and *glacial clay* valley striking ~N20°E to the south, and the small subsidiary *glacial clay* valley striking ~N50°E to the south. However, even though the main hydrostratigraphic architecture of the cognitive geological model is similar, there are still differences between the *snesim*

realizations and the cognitive geological model. The cognitive model shows clear-cut, smooth, and ordered hydrostratigraphic units. The basic modeling setup realizations reveal sporadic and random patterns. The *sand and gravel* units are placed in small lumps throughout the *glacial clay* units, but are not present within the homogenous *hemipelagic clay*. Patches of uncertain g*lacial clay* units are, however, found in the homogeneous *hemipelagic clay*, especially in the south-east corner of the Kasted survey area (Figure 9A) (Figure 10A). The same sporadic picture is seen in the vertical slices

of the realizations (Figure 9A), although, here an additional trend is revealed. The *sand and gravel* valley to the far west and at XUTM 570900 m, are not consistently filled with *sand and gravel* (Figure 9A) as in the cognitive geological model (Figure 3A). Furthermore, the EMR-map reveals that the valley margins are subject to a larger degree of ambiguity (Figure 10A), in fact at some locations the $r_{EMR}$-value is close to 1/3, which means that for the model ensemble the occurrence of either hydrostratigraphic unit is possible.

The Case 1a realizations (Figure 9B) (Figure 10B), which use the Egebjerg TI (Figure 3C), show the same overall trends as in the basic modeling setup. The subset of buried valleys mentioned above are present, however, an obvious difference is the



coarse and block-like appearance of Case 1a realization ensemble. This appearance is similar to the block-like appearance of the Egebjerg *TI* (Figure 3C). Furthermore, the horizontal slice of the realizations and EMR-map reveals that the *glacial clay* dominated area to the East has a generally larger occurrence ratio, and is thus more certain. The realizations are clearly influenced by the choice of TI, especially when Case1b is also considered (Figure 9C). The hydrostratigraphic realizations of

Case1b (Figure 9C) (Figure 10C), clearly depict the same overall buried valley trends, but the valleys in the central part of the model are largely filled with the opposite of the valley filling hydrostratigraphic units. Furthermore, the occurrence ratio seems to quite low in certain areas, such as to the south of the model, which means the ambiguity has increased. Finally, the realizations also reveal an absence of small-scale patterns, which corresponds to the conceptual TI which only contains homogenous hydrostratigraphic units.

The importance of reconstructing the incomplete resistivity grid is seen in the Case 2 (Figure 9D) (Figure 10D). The two realizations in Figure 9D show the main buried valley features, *e.g.* the western *sand and gravel* valley. However, the hydrostratigraphic units are sporadic, especially in areas with no data. Patches of *sand and gravel* and *glacial clay* are randomly spread throughout the presented horizontal cross-section and vertical slice (Figure 9D). The EMR-map also reveals an increase in low occurrence ratios in areas without soft data (Figure 10D).

The uncertainty related to the choice of geophysical modeling procedure is portrayed by Case 3. Here, *snesim* realizations are constrained to sharp resistivity models. Generally, the realizations (Figure 9E) are quite similar to the basic modeling setup realizations (Figure 9A). However, a key difference is the significant reduction or absence of patches of *glacial clay* in the homogeneous *hemipelagic clay*. In fact only one patch is found in the first realization (Figure 9E) in the south-west corner, while it is not present in the second realization, and the EMR-map further reveals a reduction of the occurrence ratios

generally, especially along the southern margin of the realizations (Figure 10E).

Case 4 shows the influence that the hard data has on the hydrostratigraphic realizations in two sub-cases: Case 4a, where *snesim* simulations are run without hard data, and Case 4b, where the borehole data is treated as soft information. Figure 9F and G shows two hydrostratigraphic realizations without hard data and with soft borehole data, respectively. These realizations do not differ significantly from the basic modeling setup realizations and in fact are quite similar. One key

difference is the central *glacial clay* valley striking ~E30°S, which does not contain any *sand and gravel* to the west (Figure 9F and G). The EMR-maps reveal that without boreholes (Figure 10F) the occurrence ratios generally decrease, making the realizations more ambiguous. The usage of the borehole data as soft information also seems to reduce the occurrence ratios compared to the basic modeling setup. Generally, leaving out the borehole data, or treating it as soft data, results in local changes in areas with a high density of boreholes.

The final case, Case 5, illustrates the importance of the SkyTEM soft data. The *snesim* simulations are run using only hard data and the cognitive geological model as a TI. The output realizations (Figure 9H) portray smooth and large-scale hydrostratigraphic units. The hydrostratigraphic architecture of the buried valleys is not reflected. However, the *sand and*





*gravel* valley, to the west, does seem to protrude slightly in the realizations (Figure 9H) and EMR-map reveals a significant decrease in the occurrence ratio, and thus an increase in the ambiguity of the model ensemble (Figure 10H).

**Figure 9:** *Each case is displayed by two realizations, realization #1 of 50, and realization #30 of 50. Each realization is portrayed as a horizontal slice at 20 mbsl, and a vertical cross-section intersecting at UTMY 6230100 m. **A** shows the realization results for the Basic modeling setup, **B** shows the realization results for Case 1a which uses the Egebjerg TI. **C** shows the realization results for the Case 1b in which the conceptual TI was used for simulation, **D** shows the results for Case 2 where the simulation is constrained to an incomplete soft data grid, **E** shows the results for Case 3 where the simulation is constrained to the sharp resistivity models, **F** shows the results for Case 4a where the simulation are run without borehole information, **G** shows the results for Case 4b where the simulations are constrained to soft borehole data, and **H** shows the results for Case 5 where the simulation is run without the soft resistivity data.*





**Figure 10:** *A presentation of the "ensemble mode ratio" (EMR) maps, computed for the different case ensembles of hydrostratigraphic models. Each EMR map is presented as a horizontal slice centered on 20 mbsl, and a vertical cross-section intersecting at UTMY 6230150m; A-H presents the EMR-type uncertainty map for each of the different cases, which are summarized in Table 1. The EMR values portray how certain the ensemble of MPS realizations are, i.e. if $r_{EMR}=1/3$ then the realization is uncertain, and we have equal probability of finding either hydrostratigraphic unit since $P(s_1)=P(s_2)=P(s_3)=33\%$. On the other hand if $r_{EMR}=1$, then each realization of the given ensemble contains the same hydrostratigraphic unit at the given grid cell.*



## 3.2. Quantitative comparison using differences in object based Euclidean Distances as a measure for similarity

The distances between each of the 400 realizations have been computed using eq. (6) and (7). The full distance matrix is presented in Figure 11A. The distances between each realization and the cognitive geological models have also been computed and plotted in Figure 11B. To aid the interpretation of the distance matrix and distances to the cognitive model a
summary table, Table 3, has been compiled.

The basic modeling setup constitutes a common *snesim* setup, with the geophysical data as soft data, boreholes as hard data, and a 3D geological conceptualization encased in a TI. The ensemble average variability is computed according to the equations presented by Barfod *et al.* (2017), and the resulting ensemble average variability is 10.1 m, with an average distance to the cognitive model of 24.3 m. This means that the Euclidean distance mismatch between the individual
realizations related to basic modeling setup is 10.1 m, and the average difference in Euclidean distance to the nearest active cell between the realizations and the cognitive model was 24.3 m.

The 3D geological conceptualization contained in the TI influences the final hydrostratigraphic realizations as illustrated in Case 1, which is divided into two sub-cases: Case 1a and Case 1b. In Case 1a, using a 3D cognitive geological model from the Egebjerg area as a TI for hydrostratigraphic simulation increases the average distance to the cognitive model to 24.9 m
(Figure 11A) (Table 3). Furthermore, the average variability has increased to 13.6 m (Figure 11B) (Table 3). The other sub-case revolves around using an entirely conceptual geological model as a TI. The conceptual TI was designed to reflect the overall geology, yet still contains some bias. The results reflect the bias, with increased distances to the cognitive geological model, which are now centered on 25.6 m (Figure 11A) (Table 3). The ensemble variability has increased to 14.8 m (Figure 11B) (Table 3).

The importance of proper reconstruction of the incomplete resistivity grid is illustrated in Case 2, where the incomplete resistivity grid was used for simulation. The resulting realization ensemble have a large ensemble variability centered on 24.1 m (Figure 11A) (Table 3). The distance to the cognitive geological model is also large, with an average value of 33.1 m (Figure 11B) (Table 3).

In Case 3 the sharp SCI models were used for simulation in place of the smooth SCI models. The realizations related to Case
3 were the closest to the cognitive model with an average value of 21 m (Figure 11B) (Table 3). The variability of Case 3 realization ensemble, *i.e.* the distances between the realizations pertaining to Case 3, is small with an average value of 9.4 m – see Table 3. Recalling the raw hydrostratigraphic realizations (Figure 9E) and the EMR-map (Figure 10E), the large reduction in distances could partly be related to the removal of non-*hemipelagic* clay units along the southern border of the model and an overall increase in confidence along the southern and southeastern border of the model.

The influence of the boreholes lithology logs on the hydrostratigraphic realizations is reflected in Case 4, which is divided into two sub-cases. In the first sub-case, Case 4a, the borehole information is not used as hard data, and the realizations are



created only using soft geophysical data and the Kasted TI. However, the borehole data is still used for creating the resistivity-hydrostratigraphic histograms (Figure 1B), which are used for creating the probability grids. The ensemble average variability is 10.7 m (Figure 11A) (Table 3) and the average distance to the cognitive model is 24.3 m (Figure 11B) (Table 3). In the second sub-case, Case 4b, the boreholes are used as soft information to reflect the uncertainty of the

5    borehole information. The ensemble average variability is 10.9 m, and the average distance to the cognitive model is 24.3 m. This illustrates how the *snesim* realizations are not particularly sensitive towards the sparse borehole hard data.

Not including the geophysical soft data in the *snesim* simulations, Case 5, resulted in the largest ensemble average variability of 40.0 m (Figure 11A) (Table 3). The average distances between Case 5 realizations and the cognitive model was 59.3 m. This means that the realizations of Case 5 are the most different from the rest of the realizations. The *snesim* realizations are

10   sensitive towards not including the geophysical data, or using the incomplete resistivity grid. This underlines the importance of the geophysical soft data in relation to hydrostratigraphic modeling using the *snesim* methodology.

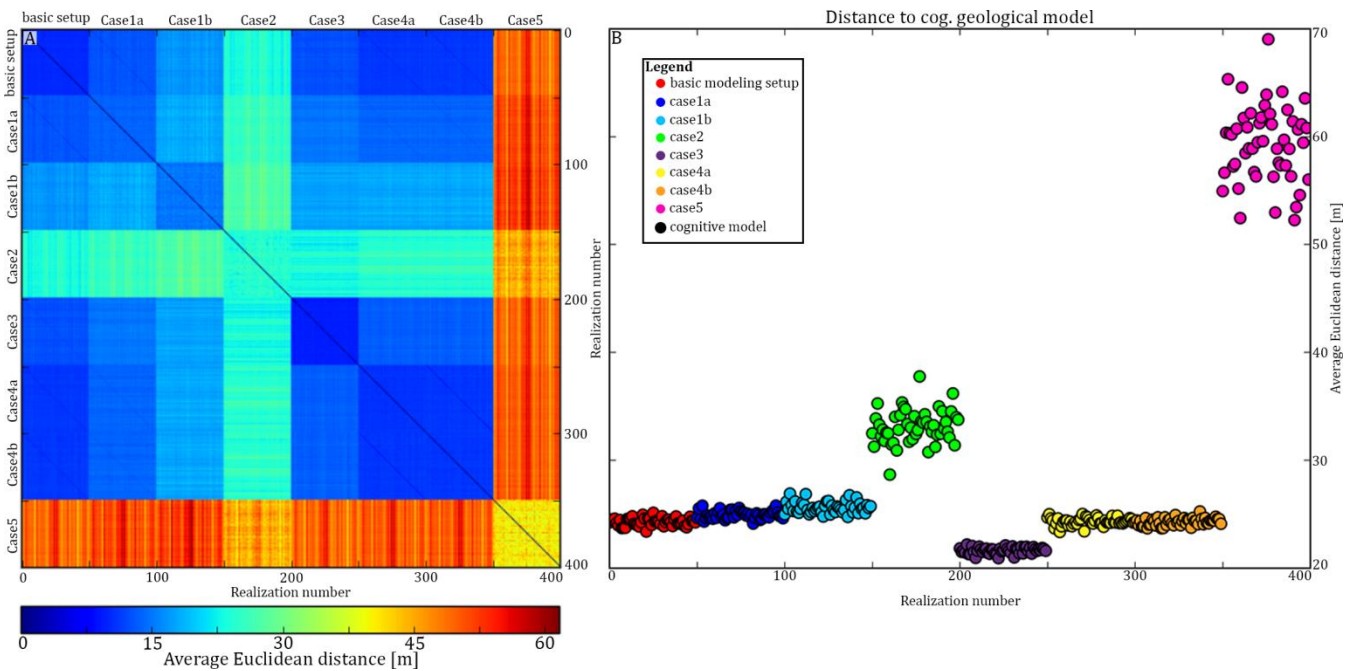

**Figure 11:** *A presentation of the average Euclidean distance calculations.* **A** *shows the full distance matrix,* **B** *shows the average Euclidean distances between each individual hydrostratigraphic realization and the cognitive geological model.*





**Table 3:** *A summary table showing the average distance value for each 50 by 50 square representing a given case in the distance matrix (Figure 11A). The final column, labelled "Distance$_{cog}$", summarizes the distances to the cognitive geological model, presented as the average of each colored point cloud in Figure 11B. The distances in parenthesis represent ensemble variabilities, and the remaining values represent average distances between different ensembles. The unit of the average distances is meters.*

| Distance [m] | Basic setup | Case1a | Case1b | Case2 | Case3 | Case4a | Case4b | Case5 | Distance$_{cog}$ |
|---|---|---|---|---|---|---|---|---|---|
| Basic setup | (10.1) | 12.9 | 16.9 | 24.0 | 12.7 | 11.1 | 11.2 | 49.6 | 24.3 |
| Case1a | 12.9 | (13.6) | 18.3 | 26.0 | 15.1 | 14.0 | 13.9 | 51.7 | 24.9 |
| Case1b | 16.9 | 18.3 | (14.8) | 27.9 | 17.8 | 18.1 | 18.1 | 52.5 | 25.6 |
| Case2 | 24.0 | 26.0 | 27.9 | (24.1) | 23.5 | 25.0 | 24.9 | 45.2 | 33.1 |
| Case3 | 12.7 | 15.1 | 17.8 | 23.5 | (9.4) | 13.6 | 13.6 | 49.6 | 21.6 |
| Case4a | 11.1 | 14.0 | 18.1 | 25.0 | 13.6 | (10.7) | 11.1 | 50.7 | 24.3 |
| Case4b | 11.2 | 13.9 | 18.1 | 24.9 | 13.6 | 11.1 | (10.9) | 50.6 | 24.3 |
| Case5 | 49.6 | 51.7 | 52.5 | 45.2 | 49.6 | 50.7 | 50.6 | (40.0) | 59.3 |

## 4 Discussion

The cognitive geological model was created based on smooth SkyTEM resistivity models and lithological logs (Høyer et al., 2015) as well as the conceptual geological understanding of the area. The model was simplified from a full 3D geological model containing a total of 42 unique geological units, to a hydrostratigraphic model containing only 3 hydrostratigraphic units. The cognitive geological model, although detailed and extensive, is not the "true" geological model. The ensemble realizations should not directly reflect the cognitive model, yet the cognitive model can be thought of as a reference point in modeling space, which we would prefer our models to resemble.

The results revealed the importance of the SkyTEM dataset. Not including the resistivity models in the MPS simulations, Case 5, yielded realizations which were both the least similar to the cognitive geological model, and with the largest variability between the individual realizations. Including the incomplete resistivity grid, Case 2, improved the realization results compared to not including them at all. Yet, the ensemble variability was large and resulting realizations were ranked second least similar to the cognitive geological model. The realization ensemble which was closest to the cognitive geological model belongs to Case 3. Here, the resistivity grid was reconstructed from the sharp SCI models, which, in this case, increase the fingerprint of resistive extreme values, which in turn results in less ambiguous reconstructed resistivity grids; compare Figure 6C and D. It should be noted that the usage of block Kriging for assigning the sharp resistivity models to the modeling grid, resulted in smoothing of sharp vertical boundaries otherwise found in sSCI models. These three cases together reveal the importance of the geophysical soft data when using the *snesim* setup presented in this study.



In relation to Case5, it can be argued that even though the SkyTEM resistivity models are not used as soft data, they are still included indirectly since the TI, or cognitive geological model, was created using smooth SkyTEM resistivity models. However, the realizations related to Case 5, revealed an ensemble of realizations which did not replicate the overall geological architecture, implying the importance of using the SkyTEM models as soft data.

On the other hand the cases related to studying the sensitivity towards borehole information, Case 4a and Case 4b, revealed that the large-scale hydrostratigraphic architecture was not changed significantly. The distance measure used in this study observes similarities or dissimilarities of large-scale hydrostratigraphic architecture, and is not sensitive towards local changes in small-scale patterns. The amount of geophysical information is relatively large, meaning the relative influence of (few) borehole data becomes less significant. This does *not* mean that the borehole data are not important; they both contain

locally accurate information and are used to estimate the regional resistivity-hydrostratigraphic relationship (Figure 1B). In other surveys, where the contrast between geophysical and lithological information is smaller, the importance of the borehole data will likely increase. In relation to this study, such small-scale changes are insignificant. Yet if the realizations are to be used for flow simulations or predictions on a smaller scale, such smaller scales might suddenly have an important impact on prediction accuracy. Additionally, if such small-scale patterns are important, the size of the model grid-cells should be

smaller to accommodate simulations of these variations. Discretizing hydrostratigraphic and groundwater models with relatively small grid-cells can be CPU demanding, depending on the total number of grid cells.

The idea of using borehole data as soft information, as seen with Case 4b, was motivated by the fact that borehole information is often considered as hard information, *i.e.* they are not associated with an uncertainty value. In this study the borehole dataset contains information from a variety of sources, and can be associated with a degree of uncertainty.

Therefore, the soft borehole probability values derived during the assigning of the boreholes to the modeling grid are combined with the SkyTEM-based probability grids using the Tau model. This approach enables the borehole probability to alter the final probability-grid, while still conditioning the SkyTEM data. Combining the information rather than letting the borehole data count as "truth", *i.e.* hard data, allows the borehole data to influence the realizations, especially if the soft borehole information disagrees with the soft geophysical data (Figure 7).

The conceptual geological understanding has always been considered an integral part of geological modeling. In this case the conceptual geological understanding is implemented via the TI, which makes it easy to alter. A total of 3 different TIs were used for simulation in this study, the Kasted, Egebjerg, and conceptual TIs. The results showed that models simulated using the Egebjerg TI, Case1a, portrayed the same overall hydrostratigraphic architecture. This opens for the possibility of using 3D cognitive geological models as TIs for new survey areas, as long as the geological settings are similar. One key

difference between the models, however, was the more block-like and coarse nature of the realizations using the Egebjerg TI, due to the coarseness of the Egebjerg TI. An important observation is that when a spatially dense and extensive geophysical dataset, such as SkyTEM, is present, the *snesim* realizations are not as sensitive towards the choice of TI, when the TI is

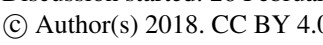



relatively similar to the expected scenario. However, as illustrated in Case 1b picking a TI which has significantly different vertical proportions (Figure 4), which do not match the soft data, the TI dominates the realizations in places where the soft data do not display a high probability for a specific hydrostratigraphic unit. In Figure 9C and Figure 10C, it can be seen that the *glacial clay valley*, both present in the soft data variable (Figure 7B) and the cognitive Kasted geological model (Figure

3A), is represented as *sand and gravel*. This leads to the conclusion that one needs to pay attention to the construction of the TI. Furthermore, the large-scale and homogenous nature of the hydrostratigraphic architecture in the conceptual TI, results in realizations which reflects the homogeneity. In comparison with the realizations based on the TIs derived from cognitive models, the realizations do not contain small-scale patterns.

The reconstruction of the resistivity grid is an important step of the *snesim* setup presented in this study. This was illustrated

in Case 2, where the incomplete resistivity grid was used instead of the reconstructed resistivity grid, resulting in larger realization variability and distance to the cognitive geological model. These realizations could have been improved by increasing the prior knowledge provided to *snesim* before simulation. One such option is to provide so-called vertical proportions, in place of solely the target global proportions. The global proportions simply give a percentage fraction of the different hydrostratigraphic units in the outcome realizations. The vertical proportions are defined for each simulation grid

layer, and determine the proportions as a function of depth. This makes sense if the different units in the realizations are clearly linked to geological units which in turn have clear stratigraphic layering. In our case, this would have impacted the realizations by not allowing the presence of *hemipelagic clay* at the top of the model. Furthermore, *sand and gravel* and *glacial clay* would not be allowed at the bottom of the model. However, vertical proportions were not used in any other cases, and were therefore not used in Case 2. The usage of vertical proportions for conditioning could also improve the

results of Case 5.

Part of the considerations of this study was to utilize the *DeeSse* code for Direct Sampling (DS) simulation. This DS implementation uses an alternative method for constraining continuous auxiliary data, presented by Chugunova and Hu (2008). The method requires the construction of multivariate TI, where a categorical TI is coupled with a continuous auxiliary variable, which is used during simulation to constrain the soft data variable. It is important that the auxiliary

variable reflects both the complications related to the given geophysical method, as well as the local petrophysical-hydrostratigraphic relations. Creating such an auxiliary variable, which spatially overlaps with the TI, and reflects the 3D AEM data set is no trivial task. The requirements for the geophysical modeling procedure are twofold. Firstly, the categorical TI needs to be populated with resistivity values, *e.g.* as in Christensen *et al.* (2017) where a Bayesian McMC algorithm is used to create 1D resistivity models drawn from a posterior probability distribution. This is no straightforward

task. Secondly, the populated resistivity model then ideally needs to be forward modelled using full 3D forward modeling code, which is computationally expensive. Alternatively, approximate 1D forward modeling is also an option. The correct system parameters of the AEM instrument and data processing paramenters have to be taken into account. Thirdly the synthetic data obtained by forward modeling must be inverted using the same procedure as the field data set. To our



knowledge, such usage of an auxiliary variable for constraining a soft geophysical data variable is not widespread within the domain of AEM geophysical methods. In this study the *snesim* method was used, which uses the τ-model (Journel, 2002). The τ-model proved a more straightforward approach when combined with the method for creating resistivity-hydrostratigraphic histograms presented by Barfod *et al.* (2016).

The study presented by Barfod *et al.* (2017) used the alternative modified Hausdorff distance (MHD) measure for comparing realizations. Due to the computational burden of the method, it was difficult to create exhaustive distance computations, *i.e.* where all information from individual realizations is used. The usage of differences in EDT of binary translations of the categorical realizations for comparing the individual realizations proved to be a more computationally feasible approach. In this paper an efficient algorithm for computing the EDT was used (Maurer et al., 2003). This computationally advantageous

approach for computing the distance between two realizations allows for a full analysis of the realizations. Each realization is then compared based on each of the hydrostratigraphic categories and on the entire 3D objects, resulting in a detailed comparison. The resulting distance matrix (Figure 11A) was able to differentiate between the realizations pertaining to the different cases. The random number seed between cases was chosen so the first realization of each case has the same random seed; the second realization has the same seed, *etc*. This can be seen in the distance matrix (Figure 11A), where off diagonal

cases have a smaller distance values along the diagonal within the given 50 by 50 sub-matrix. An example is the 50 by 50 sub-matrix between the basic setup and Case 1a, where the diagonal is clearly marked by lower distances relative to the remaining sub-matrix.

## 5 Conclusion

A hydrogeophysical data set from Kasted in Denmark was used for stochastic hydrostratigraphic simulation using the *snesim*

algorithm. The main goal of this study was to improve our understanding of ensemble hydrostratigraphic modeling variability related to stochastic *MPS* modeling. The study was divided into 8 sub-cases designed to reflect the impact related to key components of the hydrostratigraphic modeling setup, *i.e.* the TI, borehole lithology logs, and SkyTEM resistivity models. The results revealed that the hydrostratigraphic realizations were sensitive first and foremost to the geophysical dataset due to its extensive nature. Not including the geophysical data in the realizations resulted in an average Euclidean

distance variability of 40 m and a distance to the cognitive model of 59 m, which was, by far, the largest distance of all realizations. Furthermore, the geophysical modeling procedure influences the resulting realizations. It was shown that choosing so-called sharp inversion models (sSCI), in place of smooth inversion models (SCI), resulted in a realization ensemble which had similar distance based variabilities, 9.4 m and 10.1 m, respectively. However, using sSCI models decreased the distance to the cognitive geological model from 24.3 m, to 21.6 m. The choice of a TI containing a relevant

geological conceptualization is important. The cognitive Egebjerg model was used as a TI to simulate the hydrostratigraphic Kasted model, which yielded similar realizations to the case where the cognitive Kasted model was used as a TI. The



Egebjerg TI contained relevant geological architecture, but if a conceptual TI is introduced containing significantly different vertical proportions, the resulting realizations will reflect these differing vertical proportions. Finally, it was seen that the borehole lithology logs did not significantly influence the realizations. The lithology logs only carry information in the immediate vicinity of the borehole, and are sparse in comparison to the resistivity data. The boreholes therefore only have a minor influence on the realizations. The comparison measures used here mainly compare the overall large-scale architecture components of the realizations, and do not reflect small-scale changes. In relation to this study the usage of the lithology logs as hard data does not show a significant impact on the MPS realizations. However, if the hydrostratigraphic models are used for predicting groundwater flow the boreholes might be important. However, it should be mentioned that the resistivity-hydrostratigraphic histograms, which are used extensively in this research, are created from the borehole information.

## 6 Acknowledgements

I would like to thank Senior Research Engineer Celine Scheidt of Stanford University for pointing us in the direction of using Euclidean Distance Transforms for comparing MPS realizations. This study is supported by HyGEM, Integrating geophysics, geology, and hydrology for improved groundwater and environmental management, project no. 11- 116763. The funding for HyGEM is provided by The Danish Council for Strategic Research.





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





## Appendix

### A1. Simple Kriging parameters for creating borehole probability grids (Case 4)

$mean_{SK} = 1/3$

Search ellipsoid:

|        | Max | Med | Min |
|--------|-----|-----|-----|
| Ranges | 200 | 200 | 10  |
| Angles | 0   | 0   | 0   |

Variogram:

Contribution = 1

|        | Max  | Med  | Min |
|--------|------|------|-----|
| Ranges | 1000 | 1000 | 50  |
| Angles | 0    | 0    | 0   |

### A2. General SGeMS parameters used for the *snesim* realizations:

| Property name: | value/count: |
|----------------|--------------|
| algorithm name | snesim_std |
| use_pre_simulated_gridded_data | 0 |
| Use_ProbField | 1 |
| ProbField_properties | count=3, value="sg_0;ct1;pc2" |
| TauModelObject | [1 1] |
| use_vertical_proportion | 0 |
| Cmin | 5 |
| Constraint_Marginal_ADVANCED | 0 |
| resimulation_criterion | -1 |
| resimulation_iteration_nb | 1 |
| Nb_Multigrids_ADVANCED | 5 |
| Debug_Level | 0 |
| Subgrid_choice | 0 |
| expand_isotropic | 1 |
| expand_anisotropic | 0 |
| aniso_factor | NA |
| Use_Affinity | 0 |
| Use_Rotation | 0 |
| Nb_Facies | 3 |
| Marginal_Cdf | 0.19 0.24 0.57 |
| Max_Cond | 100 |
| Search_Ellipsoid | [750 750 0 0 0 0] |

Marginal cdf:

|       | *sand and gravel* | *glacial clay* | *hemipelagic clay* |
|-------|-------------------|----------------|---------------------|
| value | 0.19              | 0.24           | 0.57                |