# Peer review of "Contributions to uncertainty related to hydrostratigraphic modeling using Multiple-Point Statistics"

_Hydrology and Earth System Sciences, 2017_

## Referee Comment (RC1) · Anonymous Referee #1 · 8 Apr 2018

Review of "Contributions to uncertainty related to hydrostratigraphic modeling using Multiple-Point Statistics"

By Adrian A.S. Barfod, Troels N. Vilhelmsen, Flemming Jørgensen, Anders V. Christiansen, Julien Straubhaar and Ingelise Møller.

The main subject of the article is studying the impact of datasets used to do hydrostratigraphic modeling with the MPS framework. The authors build a "Base Case" using snesim approach, with a cognitive model as Training Image (TI), borehole data as hard data and geophysical resistivity data (SkyTEM) as soft data. Then, they present different modeling cases: using a different TI, using an incomplete resisitivity grid instead

of a full resistivity grid, using borehole data as soft data instead of hard data, inverting resistivity data with a sharp inversion model instead of a smooth inversion model. The authors assess qualitatively and quantitatively the impact of changing each of these parameters.

The main contribution of the article is the method to compare quantitatively the great number of geostatistical realizations (400). The method used is based on Analysis of Distance, with the Euclidean Distance Transform (EDT) algorithm applied to measure the "distances" between realizations. The distances serve as measures of similarity between the different cases and also between cases and the TI. According to the reviewed article, the EDT is straight forward method to assess the dissimilarity between realizations that can help in the quantification of the uncertainty of the 2D and 3D models. The smaller the distance, the more similar the realizations and thus, the smaller the impact of the changed parameter on the modeling results. It is a contribution because not many hydrogeology articles are found on the "metrics" for comparing geostatistical realizations. Plus, distance measures are discussed on MPS literature but for their use in pattern modeling from training images (Gregoire Mariethoz and Caers, 2015; Honarkhan, 2011), not for their use in uncertainty estimation. In the recent book from Mariethoz and Caers (2015), called "Multiple-point geostatistics: stochastic modeling with training images" the use of distance transforms for uncertainty purposes is not mentioned. Furthermore, in the review papers on MPS methodology, the study of the sensitivity of the model prediction to TIs and underlying datasets is suggested as an important research avenue (Hu and Chugunova, 2008).

The paper is well written, with good story-telling. Even though several cases are presented, the structure is logical and the discussion about the results of each case is clear thanks to the images presented. I would agree with the publication of this article because the method seems to be a contribution with the uncertainty appraisal of the MPS results.

Comment #1:

[Figure]

Although the proposition to use distances to assess uncertainties is interesting, it seems to me that the simple EDT is not the most adequate to capture the differences between realizations. To give one example, in figure 9 we can see that the results from "Case 1a" and "Case 1b" are very different (basically, sand to the west and clay to the east for Case 1b while Case 1a is heterogeneous in the whole model). Nevertheless, in figure 11 both cases present the same distance to the cognitive geological model. The qualitative assessment by visual means remains necessary. Did you consider using more robust methods for comparing patterns in images which take into account the positioning of the events (spatial relations) and that are less affected by scaling, rotation and translation (e.g. SIFT, IMED)?

Comment #2:

On the Kasted TI and the conceptual TI we observe channels filled with one facies, without internal variation. How come there are these intercalations of sand and clays in the simulation results?

Comment #3:

As mentioned in the discussion, the global target proportions of the units could have been replaced by the vertical proportions. It would have been interesting to see the results of these realizations with the vertical proportions, but I understand that the authors don't have them (time constraints?). What are the statistics of the results? How are those global proportions respected in each case? How the change of the parameters impacts the global target statistics?

Comment #4:

The article relies on other papers for most of the methodology, but still gives some small descriptions. Nevertheless, nothing is presented on the Direct Sampling Method used for filling the gaps on the resistivity grid. This seems to be missing in the methodology. Also, the choice of the Tau value for resistivity and boreholes (2 and 1) is not argued or

referenced. Why 2 for resistivity and 1 for boreholes?

Comment #5:

What could the authors infer about the impact of the datasets in areas where data is less dense? The study case in Denmark has good data coverage, both for geophysical surveys and for borehole data.

Technical comments:

It is not indicated what "SkyTEM" stands for.

Page 4: "Two approaches are taken"? . . . we are exçpecting a second approach

Page 7: Realizations THAT reflect the real world

Page 15 : if the grid is too sparse, then limited or no information is present which can help reconstruct missing patterns is present (repetition of "is present")

Page 29: "increased"

Page 21: comparing a "realization" (no "s")

Page 35: Journel, A. G.: "Combining Knowledge From Diverse Sources: An Alternative to Traditional Data, , 34(5), 2002". (The name of the Journal is missing, "Mathematical Geology")

---

## Referee Comment (RC2) · Anonymous Referee #2 · 11 Apr 2018

**General comments**

The manuscript "Contributions to uncertainty related to hydrostratigraphic modeling using Multiple-Point Statistics" presents an interesting study where the uncertainty related to the input data required by a multiple-point statistics (MPS) simulation framework is investigated. The research described in the manuscript, although focused on a specific case study in Denmark, could have a broader applicability and would probably be of interest for the HESS readers.

Nevertheless, I believe that the manuscript contains some major issues that should be addressed by the authors before its publication. In particular, my concerns are

related to three aspects: 1) The structure of the manuscript, 2) some missing details/discussion about important aspects of the parameterization of the methodology, 3) the way mathematical relationships are expressed.

**Specific comments**

*Manuscript structure*
A number of techniques are used within the manuscript to complete the quite complex simulation framework. Some of them are used multiple times and in different contexts (for example, the tau model). Therefore, putting their description in a separate section "Methods" would be much more helpful and would help the reader in orienting himself inside a quite complex work-flow. At the moment, the description of the methods is spreader all around the manuscript, sometimes together with the results, quite often with some repetition, which makes reading the manuscript not a smooth task. A clear example of this "breaking the rhythm" of the manuscript is for example at page 18. Also, here the description of the technique is made at the wrong place, because the method was already applied some step before in the work-flow. Another example is at page 21, where a 2D example is introduced to explain the EDT.

In addition, the comparison methods (EMR-maps,...) and the distances (EDT...) definitions would deserve a separate section, maybe just after or within the "methods" section.

There are also many locations, in particular in the "Results" sections, where too many details which would be more appropriate for the "Discussion" section are anticipated (see for example pages 27-28, lines 6, 10-11).

*Section "Basic modeling set-up"*
This section is somehow quite confusing, because the authors mix the description of the "Basic modeling set-up" with the Egedbjerg TI description. I suggest to better separate the description of the various cases.

*Tau model usage*

The tau model represents one of the crucial steps of the methodology, because it is used to take into account the soft constraints provided by geophysics, but also to combine the "borehole probability" with the SkyTEM one (Fig.7). Although some information about the tau weights are provided (i.e., in appendix), I would suggest to discuss at least briefly their choice. For example, many of the considerations made by the authors would be strongly influenced by the choice of the tau weights (see for example line 32, page 30). Some insights about the choice of these weights are provided by Allard et al (2012, DOI: 10.1007/s11004-012-9396-3). Also, what happens when the weights are $\pm\infty$? (see pp18, equation 2).

*Case studies labelling*

The provided table that summarizes all the case studies is of course useful, but overall into the manuscript (for example, in figure captions), there is very often a redundancy and some of the details of the different methods, which are repeated multiple times. Maybe you should reference much more often to Table 1 and to the "codes" like "Case 1a", "Case 1b" only, and avoid repeating the detailed differences. One example of these repetition can be observed in Figure captions (see for example Fig.9, page 25).

*Introduction* pp4, lines 20-

Here I would also mention the problems related to the solution of the inverse problem (IP) in itself. By the way, this also somehow motivates your efforts in trying two different inversion techniques, like SCI and sSCI.

pp5, lines 27-

Here I believe you are already providing too much details for an introduction.

*Mathematical formulation*

The mathematical formulation is often cumbersome, because very often long text lines are used to define quantities and as subscripts. I strongly suggest to lighten the notation avoiding long text lines, and using the many letters provided by the alphabets. For

example, $N$ and $M$ could be used instead of $N_{\text{realizations}}$ and $N_{\text{cells}}$; another example is the definition of $\mathbf{D}_{i,j}$ (page 22). In addition, some relationship could be condensed and generalized. In this way, they could be written only once and contribute to shorten the manuscript. See for example (7) and (8) at page 21.

*TI non-stationarity*
In section 3.1 but also in other parts of the manuscript the imprecision of MPS in re-producing some features is clearly depicted. However, I believe that many of the en-countered problems are due to the non-stationarity of the used TI. Therefore, although of course taking into account for the geophysics helps, I would suggests to at least mention and briefly describe the role of the non-stationarity of the TIs.

*Variograms*
Could you briefly mention which variogram model you used for example to create the borehole footprint (pp18, lines 3-6)?

*Figures*
Fig.4: Please add a comment related to the spatial scale of the Egebjer TI, which is quite different from the other two. Also, it would be quite nice to add a sub-figure containing the same vertical proportions for the borehole logs.

Fig.11: The label "Realization number" in the vertical axis of part A is too close to part B and in therefore misleading. Also, I believe that the results of part B could be condensed using box-plots, one box-plot for each case. In this way, the fictitious and misleading order of the "realization number" would be by-passed.

**Technical corrections**

pp6, line 9
Please check the order in "33 line km spatially..."

pp8, line 30
It looks like the reference to Fig.4 is missing between Fig.3 and Fig.5.

pp9, line 5
"data is" => "data are"

pp18, line 18
"[2,1]" is somehow confusing with the index that you introduce some equations before...
I would specify that they are float values, writing explicitly 2.0 or 1.0.

pp20, line 25
Maybe "(3)" => "(4)"?

pp21, equation (6)
Please check for the missing $i$ subscript to $v$

pp21, line 20
Here Delta appears in the formula, but not in the following text... please check.

pp22, line17
"realizations using" => "realizations computed using" (?)

pp27, lines 6-11, 30-31
This is somehow repetitive. Please try to avoid repetitions also in other locations in the text, but in particular in this section.

pp30, line 26
"to alter...?"

---

## Referee Comment (RC3) · Anonymous Referee #3 · 13 Apr 2018

This paper is the product of a nice piece of work and is very interesting. My main concerns are related to the introduction, where some transitions and justifications are missing and to the methodological section, that goes too fast into details. I therefore suggest the following minor revisions.

In the introduction, transitions are often missing between paragraphs, for instance, page 2, between lines 14 and 15. It is a bit jumps from one topic to another.

Related to the paragraph comprised between lines 3 and 14, you might cite the following paper that discuss uncertainty and bias in training images : Ferré, Ty. "Revisiting the Relationship Between Data, Models, and Decision-Making." Groundwater 55, no.

[Figure]

5 (2017): 604-614. Pirot, Guillaume. "Using training images to build model ensembles with structural variability." Groundwater 55, no. 5 (2017): 656-659.

General justifications are given in the first paragraph of the introduction, but the authors should also justify why they chose this specific sites, and what they bring or want to improve, with regards to previous studies conducted at the Kasted site.

You should also define your notions of hard and soft conditioning clearly in the introduction

Then, in section 2, when presenting the study area, the historic of previous research could be explained/clarified. Page 6, line 5: one 'complex' too much? Page 6, line 14: how is defined the 'selected quality threshold?'

The main aspect, regarding the method subsection is that the reader is lost in details from the beginning. A big picture of the approach is missing. In the way of presenting, I would recommend to give first an overview of the method, and progressively go into details.

Page 7, lines 7 to 15 are not very clear. Can you reformulate? Page 8, line 26, figure 5 is called, while figure 4 has not been called.

---

## Author Comment (AC1) · 21 May 2018

**Response to referees, hess-2017-734**
* * *
Firstly, the authors would like to thank the three anonymous referees for taking their time to read the manuscript and providing detailed and constructive comments. The comments, questions and suggestions are addressed in the following response.
* * *
**\*RC: Referee Comments**
**\*AC:** Author Comments

**Response to anonymous referee #1**

**General comments:**
'*The main subject of the article is studying the impact of datasets used to do hydrostratigraphic modeling with the MPS framework. The authors build a "Base Case" using snesim approach, with a cognitive model as Training Image (TI), borehole data as hard data and geophysical resistivity data (SkyTEM) as soft data. Then, they present different modeling cases: using a different TI, using an incomplete resisitivity grid instead of a full resistivity grid, using borehole data as soft data instead of hard data, inverting resistivity data with a sharp inversion model instead of a smooth inversion model. The authors assess qualitatively and quantitatively the impact of changing each of these parameters. The main contribution of the article is the method to compare quantitatively the great number of geostatistical realizations (400). The method used is based on Analysis of Distance, with the Euclidean Distance Transform (EDT) algorithm applied to measure the "distances" between realizations. The distances serve as measures of similarity between the different cases and also between cases and the TI. According to the reviewed article, the EDT is straight forward method to assess the dissimilarity between realizations that can help in the quantification of the uncertainty of the 2D and 3D models. The smaller the distance, the more similar the realizations and thus, the smaller the impact of the changed parameter on the modeling results. It is a contribution because not many hydrogeology articles are found on the "metrics" for comparing geostatistical realizations. Plus, distance measures are discussed on MPS literature but for their use in pattern modeling from training images (Gregoire Mariethoz and Caers, 2015; Honarkhan, 2011), not for their use in uncertainty estimation. In the recent book from Mariethoz and Caers (2015), called "Multiple-point geostatistics: stochastic modeling with training images" the use of distance transforms for uncertainty purposes is not mentioned. Furthermore, in the review papers on MPS methodology, the study of the sensitivity of the model prediction to TIs and underlying datasets is suggested as an important research avenue (Hu and Chugunova, 2008).*
*The paper is well written, with good story-telling. Even though several cases are presented, the structure is logical and the discussion about the results of each case is clear thanks to the images presented. I would agree with the publication of this article because the method seems to be a contribution with the uncertainty appraisal of the MPS results.*'

**RC1:** '*Although the proposition to use distances to assess uncertainties is interesting, it seems to me that the simple EDT is not the most adequate to capture the differences between realizations. To give one example, in figure 9 we can see that the results from "Case 1a" and "Case 1b" are very different (basically, sand to the west and clay to the east for Case 1b while Case 1a is heterogeneous in the whole model). Nevertheless, in figure 11 both cases present the same distance to the cognitive geological model. The qualitative assessment by visual*

*means remains necessary. Did you consider using more robust methods for comparing patterns in images which take into account the positioning of the events (spatial relations) and that are less affected by scaling, rotation and translation (e.g. SIFT, IMED)?*'

**AC1:** The usage of the simple EDT for computation of the 'distance' between different realizations is a computationally feasible method for comparing 400 realizations each containing 229*133*39 cells (1,187,823 cells). The EDT method was therefore used because it thoroughly compares the average mismatch between the different realizations. However, we agree that the method is a bit simplistic for comparing heterogeneous geological objects. Other methods were in fact considered, but the EDT-based approach was the most suitable for getting started with research into comparing stochastic MPS modelling. It would be interesting to study, but was left out of the paper to keep focus on the large number of cases and assessment of the uncertainty related to them.

Changes to the manuscript: We will mention that other methods for comparing realizations exist in the introduction on page 5 lines 21-32 and add a few sentences more in the discussion section on distance measures on page 32 lines 5-17.

**RC2:** '*On the Kasted TI and the conceptual TI we observe channels filled with one facies, without internal variation. How come there are these intercalations of sand and clays in the simulation results?*'

**AC2:** This is due to the fact that the simulations are probabilistic in nature, and are based on random processes. At the beginning of the simulation a random path is drawn so that the simulation grid is filled by visiting each grid cell only once. The fine-scale patterns are partly due to the hard data which are inserted into the simulation grid before the simulation commences, and is excluded from the simulation path. As the grid is filled out, the hard borehole data might suggest a certain category but the soft data suggests another category. As the grid is filled out the overall category from the soft data dominates and if the random path visits the grid cells near the hard data point towards the end of the random path, then we are left with a small intercolation (e.g. Hansen et al., 2018). The intercolations are also inherent in the simulations without hard data, and is mainly due to process randomness related to how the snesim algorithm draws from a cumulative density function (CDF) (Strebelle, 2002)

Changes to the manuscript: The above comment did not result in any changes to the manuscript.

**RC3:** '*As mentioned in the discussion, the global target proportions of the units could have been replaced by the vertical proportions. It would have been interesting to see the results of these realizations with the vertical proportions, but I understand that the authors don't have them (time constraints?). What are the statistics of the results? How are those global proportions respected in each case? How the change of the parameters impacts the global target statistics?*'

**AC3:** The simulations have not been run with the vertical proportions since this would constitute an entire set of cases on its own, and due to the length of the paper and time constraints this was not considered. Additionally, regarding the compilation of the global proportion statistics, again, this would increase the length of the paper and shift the focus away from the main topic which is using distance based similarity measures for comparing a large collection of MPS realizations.

Changes to the manuscript: Due to the general length of the paper, we would like to avoid adding an extra case and expanding the paper to contain the global proportions statistics for each realization, unless the global statistics reveal some significant patterns relevant to the research topic of comparing MPS realizations using a 'distance' measures.

**RC4:** '*The article relies on other papers for most of the methodology, but still gives some small descriptions. Nevertheless, nothing is presented on the Direct Sampling Method used for filling the gaps on the resistivity grid. This seems to be missing in the methodology. Also, the choice of the Tau value for resistivity and boreholes (2 and 1) is not argued or referenced. Why 2 for resistivity and 1 for boreholes?*'

**AC4:** We see your points. The choice of Tau model was purely based on a series of tests, which are not mentioned and should be mentioned. Different combinations of Tau values were exhaustively tested, and the chosen values resulted were chosen since they resulted in simulations in which a smooth transition between borehole conditioned areas and non-borehole conditioned areas was seen.

Changes to the manuscript: A presentation on the Direct Sampling Method used for filling the gaps in the resistivity grid will be added as a section in 2 Materials and methods. It will be added to the text that the choice of Tau values are based on exhaustive tests.

**RC5:** '*What could the authors infer about the impact of the datasets in areas where data is less dense? The study case in Denmark has good data coverage, both for geophysical surveys and for borehole data.*'

**AC5:** When the dataset is less dense than the Kasted dataset, the simulations have less soft data for conditioning. The less conditioning data which is available, the more the simulation relies on the TI for conditioning. An extreme of this scenario is when no conditioning data is used, and the realization is entirely unconditioned (e.g. Strebelle and Journel, 2000). Therefore, the choice of TI would impact the simulation result in a larger degree if less conditioning data is present.

Changes to the manuscript: A description of the above might be added to the manuscript, provided a logical place is found.

**Technical comments:**
- It is not indicated what "SkyTEM" stands for.
  - It is a system name for an airborne transient electromagnetic system, which will be mentioned when it is first used in the abstract.
- Page 4: "Two approaches are taken"? … we are expecting a second approach…
  - Yes, the other common approach should be included in this paragraph.
- Page 7: Realizations THAT reflect the real world
  - P. 7 line 15: this will be corrected in the revised paper.
- Page 15: if the grid is too sparse, then limited or no information is present which can help reconstruct missing patterns is present (repetition of "is present")
  - P. 15 line 9: this will be corrected in the revised paper.
- Page 29: "increased"

- P. 29 line 18: this will be corrected in the revised paper.
- Page 21: comparing a "realization" (no "s")
  - P. 21 line 26: this will be corrected in the revised paper.
- Page 35: Journel, A. G.: "Combining Knowledge From Diverse Sources: An Alternative to Traditional Data, , 34(5), 2002". (The name of the Journal is missing, "Mathematical Geology")
  - The mentioned reference will correctly contain the journal name in the revised paper.

**Response to anonymous referee #2**

**General comments:**
'*The manuscript "Contributions to uncertainty related to hydrostratigraphic modeling using Multiple-Point Statistics" presents an interesting study where the uncertainty related to the input data required by a multiple-point statistics (MPS) simulation framework is investigated. The research described in the manuscript, although focused on a specific case study in Denmark, could have a broader applicability and would probably be of interest for the HESS readers.*
*Nevertheless, I believe that the manuscript contains some major issues that should be addressed by the authors before its publication. In particular, my concerns are related to three aspects: 1) The structure of the manuscript, 2) some missing details/discussion about important aspects of the parameterization of the methodology, 3) the way mathematical relationships are expressed.*'

**RC1:** '*Manuscript structure: A number of techniques are used within the manuscript to complete the quite complex simulation framework. Some of them are used multiple times and in different contexts (for example, the tau model). Therefore, putting their description in a separate section "Methods" would be much more helpful and would help the reader in orienting himself inside a quite complex work-flow. At the moment, the description of the methods is spreader all around the manuscript, sometimes together with the results, quite often with some repetition, which makes reading the manuscript not a smooth task. A clear example of this "breaking the rhythm" of the manuscript is for example at page 18. Also, here the description of the technique is made at the wrong place, because the method was already applied some step before in the work-flow. Another example is at page 21, where a 2D example is introduced to explain the EDT.*
*In addition, the comparison methods (EMR-maps,...) and the distances (EDT...) definitions would deserve a separate section, maybe just after or within the "methods" section. There are also many locations, in particular in the "Results" sections, where too many details which would be more appropriate for the "Discussion" section are anticipated (see for example pages 27-28, lines 6, 10-11).*'

**AC1:** Under the preparation of the manuscript we have worked iteratively on the structure of the manuscript and ended up with this structure as the most reader friendly consisting of a large Material and methods section (section 2), divided into the study area (2.1 The kasted study area), a general method description section (2.2 Multiple-Point Statistics (MPS) and single normal equation simulation (snesim)), a detailed description of the MPS modeling set-up with the methods related to each case (2.3 MPS modeling setup) and lastly a section of the methods used for comparing the simulation results. Generally, the style chosen for this manuscript was to explain some of the methods utilized in the, as mentioned, quite complex simulation framework was to use practical examples. This often aids the reader with figures and a purpose for applying the given method.

For instance, the usage of the Tau model is used in different contexts and is clearly explained in section "2.3.5 Case 4 – borehole lithology logs". It therefore seems unwarranted to create a separate section, which repeats the entire description of the Tau model. Alternatively, the detailed description of the Tau model could be removed from section "2.3.5 Case 4 – borehole lithology logs", but the authors prefer to present the Tau model with a specific case in mind. In this scenario we used case 4 to describe the usage of the Tau model, since the reader would be able to use Figure 7 as a visual aid in understanding the Tau method.
Changes to the manuscript: The method descriptions which are not found in section "2 Materials and Methods", but instead are found in section "3 Results", as mentioned by Anonymous referee #2, should be moved to section "2 Materials and Methods". However, brief descriptions which are used to help the reader,

e.g. P. 23 lines 14-15 where the EMR maps are briefly explained again, should not be moved to section "2 Materials and Methods".

**RC2:** *'Section "Basic modeling set-up": This section is somehow quite confusing, because the authors mix the description of the "Basic modeling set-up" with the Egebjerg TI description. I suggest to better separate the description of the various cases.'*

**AC2:** This was a mistake. The reference to Table 2 for some reason included text from the section above or below Table 2.

Changes to the manuscript: The Egebjerg TI description will solely be placed in section 2.32 Case 1 – Conceptual geological understanding.

**RC3:** *'Tau model usage: The tau model represents one of the crucial steps of the methodology, because it is used to take into account the soft constraints provided by geophysics, but also to combine the "borehole probability" with the SkyTEM one (Fig.7). Although some information about the tau weights are provided (i.e., in appendix), I would suggest to discuss at least briefly their choice. For example, many of the considerations made by the authors would be strongly influenced by the choice of the tau weights (see for example line 32, page 30). Some insights about the choice of these weights are provided by Allard et al (2012, DOI: 10.1007/s11004-012-9396-3). Also, what happens when the weights are_1? (see pp18, equation 2).'*

**AC3:** Here we refer to the Author Comment 4 (AC4) in the response to anonymous referee #1. Briefly, in relation to the choice of Tau parameters for the Tau model it should be mentioned that a series of tests have actually been carried out to select the final Tau values.

Changes to the manuscript: It will be added to the text that the choice of Tau values are based on exhaustive tests.

**RC4:** *'Case studies labelling: The provided table that summarizes all the case studies is of course useful, but overall into the manuscript (for example, in figure captions), there is very often a redundancy and some of the details of the different methods, which are repeated multiple times.*
*Maybe you should reference much more often to Table 1 and to the "codes" like "Case 1a", "Case 1b" only, and avoid repeating the detailed differences. One example of these repetition can be observed in Figure captions (see for example Fig.9, page 25).'*

**AC4:** This would certainly make the paper more concise in certain parts. However, it would also require the reader to constantly avert his/her attention to Table 1.

Changes to the manuscript:  In the revision of the manuscript Table 1 will be referenced more often to avoid repetition, to the extent we find reasonable without disturbing the reading experience.

**RC5:** '*Introduction pp4, lines 20-XX: Here I would also mention the problems related to the solution of the inverse problem (IP) in itself. By the way, this also somehow motivates your efforts in trying two different inversion techniques, like SCI and sSCI.*'

**AC5:** This is a good point.

Changes to the manuscript: A brief description of the problems related to the inversion process will be added to the introduction.

**RC6:** '*pp5, lines 27-: Here I believe you are already providing too much details for an introduction.*'

**AC6:** Okay

Changes to the manuscript: we will try to provide less details in the Introduction in the revised paper.

**RC7:** '*Mathematical formulation: The mathematical formulation is often cumbersome, because very often long text lines are used to define quantities and as subscripts. I strongly suggest to lighten the notation avoiding long text lines, and using the many letters provided by the alphabets. For example, N and M could be used instead of Nrealizations and Ncells; another example is the definition of Di;j (page 22). In addition, some relationship could be condensed and generalized. In this way, they could be written only once and contribute to shorten the manuscript. See for example (7) and (8) at page 21.*'

**AC7:** We agree that the mathematical formulations could be written in a more concise manner. However, regarding eq. (7) and (8), again, we believe it aids the reader to first see the formula with a specific case in mind and then generalize afterwards.

Changes to the manuscript: The mathematical formulas will generally be shortened and simplified for the revised paper.

**RC8:** '*TI non-stationarity: In section 3.1 but also in other parts of the manuscript the imprecision of MPS in reproducing some features is clearly depicted. However, I believe that many of the encountered problems are due to the non-stationarity of the used TI. Therefore, although of course taking into account for the geophysics helps, I would suggest to at least mention and briefly describe the role of the non-stationarity of the TIs.*'

**AC8:** The geophysical data is so spatially dense that TI non-stationarity is not as big of an issue as when less conditioning data is present. We therefore decided to not include this complication, to not confuse the reader. However, it might be wise to mention that this is the case and that even though the Tis are non-stationary, the spatially dense geophysical data actually helps in the matter.

Changes to the manuscript: Where appropriate, we will add sentences on non-stationarity of the TIs'.

**RC9:** '*Variograms: Could you briefly mention which variogram model you used for example to create the borehole footprint (pp18, lines 3-6)?*'

**AC9:** It is mentioned both in the text where you indicated, but also in the Appendix under section "A1".

Changes to the manuscript: We will add the variogram model type, e.g. exponential, spherical, etc., to the Appendix under section "A1".

**RC10:** '*Fig.4: Please add a comment related to the spatial scale of the Egebjerg TI, which is quite different from the other two. Also, it would be quite nice to add a sub-figure containing the same vertical proportions for the borehole logs.*'

**AC10:** That is a good point!

Changes to the manuscript: A comment will be made regarding the fact that the ´vertical scale of the Egebjerg model is quite different from the other Tis. Provided the creation of a sub-figure containing the borehole vertical proportions is not too time consuming, we will implement this sub-figure.

**RC11:** '*Fig.11: The label "Realization number" in the vertical axis of part A is too close to part B and is therefore misleading. Also, I believe that the results of part B could be condensed using box-plots, one box-plot for each case. In this way, the fictitious and misleading order of the "realization number" would be by-passed.*'

**AC11:** Yes, the Fig.11B of the figure will be given a bit more space so that it is clear that the "Realization number" is part of Fig.11A. We would prefer to not use boxplots since they mainly provide summary statistics and we would prefer that the reader can see the actual distance values instead.

**Technical comments:**
- pp6, line 9: Please check the order in "33 line km spatially..."
  - This will be re-ordered correctly.
- pp8, line 30: It looks like the reference to Fig.4 is missing between Fig.3 and Fig.5.
  - We will add a sentence with reference to Fig. 4, so it will be mentioned before Fig. 5
- pp9, line 5: "data is" => "data are"
  - This will be corrected in the revised paper.
- pp18, line 18: "[2,1]" is somehow confusing with the index that you introduce some equations before... I would specify that they are float values, writing explicitly 2.0 or 1.0.
  - This will be added to the revised manuscript.
- pp20, line 25: Maybe "(3)" => "(4)"?
  - This will be corrected.
- pp21, equation (6): Please check for the missing i subscript to v
  - There is no missing subscript. The "$u$" symbol represents a single location vector, while the "$v$" is a set of vectors contained by "V". Therefore, the formula only shows the computation of the $d_{EDT}$ at a single location. The process must be repeated for all points for the grid. Perhaps this should be mentioned in the revised paper.
- pp21, line 20: Here Delta appears in the formula, but not in the following text... please check.
  - This will be corrected in the revised paper.

- pp22, line17: "realizations using" => "realizations computed using" (?)
  - This will be corrected in the revised paper.
- pp27, lines 6-11, 30-31: This is somehow repetitive. Please try to avoid repetitions also in other locations in the text, but in particular in this section.
  - We couldn't find the particular repetition mentioned here, but will try to remove as many of such repetitions in the revised paper.
- pp30, line 26: "to alter...?"
  - Will be corrected, however, it seems like it is a logical statement.

**Response to anonymous referee #3**

**General comments:**
*'This paper is the product of a nice piece of work and is very interesting. My main concerns are related to the introduction, where some transitions and justifications are missing and to the methodological section, that goes too fast into details. I therefore suggest the following minor revisions.'*

**RC1:** *'In the introduction, transitions are often missing between paragraphs, for instance, page 2, between lines 14 and 15. It is a bit jumps from one topic to another. Related to the paragraph comprised between lines 3 and 14, you might cite the following paper that discuss uncertainty and bias in training images : Ferré, Ty. "Revisiting the Relationship Between Data, Models, and Decision-Making." Groundwater 55, no. 5 (2017): 604-614. Pirot, Guillaume. "Using training images to build model ensembles with structural variability." Groundwater 55, no. 5 (2017): 656-659.'*

**AC1:** Good points.

Changes to the manuscript: The introduction will be cleaned up so that transitions between paragraphs are less abrupt in the revised paper. The two references will also be added in the revised paper.

**RC2:** *'General justifications are given in the first paragraph of the introduction, but the authors should also justify why they chose this specific sites, and what they bring or want to improve, with regards to previous studies conducted at the Kasted site.'*

**AC2:** That is a good point. The main justification for using the Kasted site was that the geology of the area was fairly simple, thus the dataset is also, to a degree, simple. Furthermore, a geological model of the area had already been compiled, meaning we already had information about the hydrogeology.

Changes to the manuscript: The justification for using the Kasted dataset should be mentioned in the introduction and will be added. We will also revise the paper so that it reflects what we hope to improve in regards to the Kasted model.

**RC3:** *'You should also define your notions of hard and soft conditioning clearly in the introduction'*

**AC3:** Okay.

Changes to the manuscript: We will add this definition in the revised paper.

**RC4:** *'Then, in section 2, when presenting the study area, the historic of previous research could be explained/clarified. Page 6, line 5: one 'complex' too much? Page 6, line 14: how is defined the 'selected quality threshold?''. '*

**AC4:** If anonymous referee #3 is referring to the fact that we should explain previous research in the Kasted area, then the main research which has been conducted in the area is compiling the cognitive 3D geological model of the area. Therefore, we do not desire going into more detail.

Changes to the manuscript: The poor usage of the word 'complex' will be corrected. A reference to the borehole quality assessment method used will be added here and the method is briefly introduced.

**RC5:** '*The main aspect, regarding the method subsection is that the reader is lost in details from the beginning. A big picture of the approach is missing. In the way of presenting, I would recommend to give first an overview of the method, and progressively go into details.*'

**AC5:** Perhaps it would be wise to include an overview of what we hope to achieve by stochastic modelling.

**RC6:** '*Page 7, lines 7 to 15 are not very clear. Can you reformulate?*'

**AC6:** Will be reformulated in the revised paper.

**RC7:** '*Page 8, line 26, figure 5 is called, while figure 4 has not been called.*'

**AC7:** We will fix this by adding a reference to Figure 4 before we call Figure 5.

**References**

Hansen, T.M., Vu, L.T., Mosegaard, K., Cordua, K.S., 2018. Multiple point statistical simulation using uncertain (soft) conditional data. Comput. Geosci. 114, 1–10. https://doi.org/10.1016/j.cageo.2018.01.017

Strebelle, S., Journel, A.G., 2000. Sequential simulation drawing structures from training images. Stanford University.

Strebelle, S.: Conditional Simulation of Complex Geological Structures Using Multiple-Point Statistics, Math. Geol., 34(1), 1–21, doi:10.1023/A:1014009426274, 2002.

---

## Author Response (AR1)

Dear Mr. Carrera,

We have decided to compile a revised version of our paper according to the comments from the three anonymous reviewers.

As previously discussed we have added co-author Anne-Sophie Høyer to the author list, since she was somehow forgotten.

In particular, we have rearranged the paper so that a dedicated methods section is now present, which focuses on introducing the different methods, such as the Tau model, MPS, and distance methods. This resulted in the methods being introduced before they are used. The mathematical formulations have been cleaned up and shortened. Generally, the referee comments have been addressed and following you should find our response and a complete list of all corrections.

Best,
Adrian S. Barfod

**Response to referees, hess-2017-734**

Firstly, the authors would like to thank the three anonymous referees for taking their time to read the manuscript and providing detailed and constructive comments. The comments, questions and suggestions are addressed in the following response.

**\*RC: Referee Comments**
**\*AC:** Author Comments

*NOTE: All line numbers in this document refer to the revised paper and not the "manuscript changes document" or any previously submitted versions of this paper.*

**Response to anonymous referee #1**

**General comments:**
*'The main subject of the article is studying the impact of datasets used to do hydrostratigraphic modeling with the MPS framework. The authors build a "Base Case" using snesim approach, with a cognitive model as Training Image (TI), borehole data as hard data and geophysical resistivity data (SkyTEM) as soft data. Then, they present different modeling cases: using a different TI, using an incomplete resisitivity grid instead of a full resistivity grid, using borehole data as soft data instead of hard data, inverting resistivity data with a sharp inversion model instead of a smooth inversion model. The authors assess qualitatively and quantitatively the impact of changing each of these parameters. The main contribution of the article is the method to compare quantitatively the great number of geostatistical realizations (400). The method used is based on Analysis of Distance, with the Euclidean Distance Transform (EDT) algorithm applied to measure the "distances" between realizations. The distances serve as measures of similarity between the different cases and also between cases and the TI. According to the reviewed article, the EDT is straight forward method to assess the dissimilarity between realizations that can help in the quantification of the uncertainty of the 2D and 3D models. The smaller the distance, the more similar the realizations and thus, the smaller the impact of the changed parameter on the modeling results. It is a contribution because not many hydrogeology articles are found on the "metrics" for comparing geostatistical realizations. Plus, distance measures are discussed on MPS literature but for their use in pattern modeling from training images (Gregoire Mariethoz and Caers, 2015; Honarkhan, 2011), not for their use in uncertainty estimation. In the recent book from Mariethoz and Caers (2015), called "Multiple-point geostatistics: stochastic modeling with training images" the use of distance transforms for uncertainty purposes is not mentioned. Furthermore, in the review papers on MPS methodology, the study of the sensitivity of the model prediction to TIs and underlying datasets is suggested as an important research avenue (Hu and Chugunova, 2008).*
*The paper is well written, with good story-telling. Even though several cases are presented, the structure is logical and the discussion about the results of each case is clear thanks to the images presented. I would agree with the publication of this article because the method seems to be a contribution with the uncertainty appraisal of the MPS results.'*

**RC1:** *'Although the proposition to use distances to assess uncertainties is interesting, it seems to me that the simple EDT is not the most adequate to capture the differences between realizations. To give one example, in*

*figure 9 we can see that the results from "Case 1a" and "Case 1b" are very different (basically, sand to the west and clay to the east for Case 1b while Case 1a is heterogeneous in the whole model). Nevertheless, in figure 11 both cases present the same distance to the cognitive geological model. The qualitative assessment by visual means remains necessary. Did you consider using more robust methods for comparing patterns in images which take into account the positioning of the events (spatial relations) and that are less affected by scaling, rotation and translation (e.g. SIFT, IMED)?'*

**AC1:** The usage of the simple EDT for computation of the 'distance' between different realizations is a computationally feasible method for comparing 400 realizations each containing 229*133*39 cells (1,187,823 cells). The EDT method was therefore used because it thoroughly compares the average mismatch between the different realizations. However, we agree that the method is a bit simplistic for comparing heterogeneous geological objects. Other methods were in fact considered, but the EDT-based approach was the most suitable for getting started with research into comparing stochastic MPS modelling. It would be interesting to study, but was left out of the paper to keep focus on the large number of cases and assessment of the uncertainty related to them

**Changes to the manuscript:**
Lines 141-144: the IMED and SIFT distances are mentioned as alternative methods for comparing 3D hydrostratigraphic MPS realizations.

**RC2:** *'On the Kasted TI and the conceptual TI we observe channels filled with one facies, without internal variation. How come there are these intercalations of sand and clays in the simulation results?'*

**AC2:** This is due to the fact that the simulations are probabilistic in nature, and are based on random processes. At the beginning of the simulation a random path is drawn so that the simulation grid is filled by visiting each grid cell only once. The fine-scale patterns are partly due to the hard data which are inserted into the simulation grid before the simulation commences, and is excluded from the simulation path. As the grid is filled out, the hard borehole data might suggest a certain category but the soft data suggests another category. As the grid is filled out the overall category from the soft data dominates and if the random path visits the grid cells near the hard data point towards the end of the random path, then we are left with a small intercolation (Hansen et al., 2018). The Intercolations are also inherent in the simulations without hard data, and is mainly due to process randomness related to how the snesim algorithm draws from a cumulative density function (CDF) (REF: Strebelle 2002, snesim paper)

**Changes to the manuscript:**
The above comment did not result in any changes to the manuscript.

**RC3:** *'As mentioned in the discussion, the global target proportions of the units could have been replaced by the vertical proportions. It would have been interesting to see the results of these realizations with the vertical proportions, but I understand that the authors don't have them (time constraints?). What are the statistics of the results? How are those global proportions respected in each case? How the change of the parameters impacts the global target statistics?'*

**AC3:** The simulations have not been run with the vertical proportions since this would constitute an entire set of cases on its own, and due to the length of the paper and time constraints this was not considered. Regarding the compilation of the global proportions statistics, this has not been reported since the focus of this paper was to compare the results based on distance based similarities. However, provided it is not too time consuming it could still be a possibility to report the average global proportion statistics for the different cases.

**Changes to the manuscript:**
Due to the general length of the paper, we would like to avoid adding an extra case and expanding the paper to contain the global proportions statistics for each realization, therefore we have made no changes to the manuscript.

**RC4:** '*The article relies on other papers for most of the methodology, but still gives some small descriptions. Nevertheless, nothing is presented on the Direct Sampling Method used for filling the gaps on the resistivity grid. This seems to be missing in the methodology. Also, the choice of the Tau value for resistivity and boreholes (2 and 1) is not argued or referenced. Why 2 for resistivity and 1 for boreholes?*'

**AC4:** We see your points. The choice of Tau model was purely based on a series of tests, which are not mentioned and should be mentioned. Different combinations of Tau values were exhaustively tested, and the chosen values resulted were chosen since they resulted in simulations in which a smooth transition between borehole conditioned areas and non-borehole conditioned areas was seen.

**Changes to the manuscript:**
Lines 215-231: A dedicated section, section 3.2, has been added describing the usage of Direct Sampling Method for reconstructing incomplete geophysical datasets
Lines 193-214: A dedicated section, section 3.1.1, introducing the Tau model has been added
Lines 508-511: The exhaustive tests for choosing the tau model parameters are described

**RC5:** '*What could the authors infer about the impact of the datasets in areas where data is less dense? The study case in Denmark has good data coverage, both for geophysical surveys and for borehole data.*'

**AC5:** When the dataset is less dense than the Kasted dataset, the simulations have less soft data for conditioning. The less conditioning data which is available, the more the simulation relies on the TI for conditioning. An extreme of this scenario is when no conditioning data is used, and the realization is entirely unconditioned (e.g. Strebelle and Journel, 2000). Therefore, the choice of TI would impact the simulation result in a larger degree if less conditioning data is present.

**Changes to the manuscript:**
Lines 697-701: A sections discussing the good data coverage and non-stationarity has been added

**Technical comments:**
- It is not indicated what "SkyTEM" stands for.

**Author response:** It is a system name for an airborne transient electromagnetic system, i.e. the name of a company.

**Resulting changes:**
Line 54: the sentence is changed to "The airborne transient electromagnetic SkyTEM data …"

- Page 4: "Two approaches are taken"? … we are expecting a second approach…
    **Author response:** Yes, the other common approach should be included in this paragraph.
    **Resulting changes:** Lines 221-223: the other common approach is described

- Page 7: Realizations THAT reflect the real world
    **Author response:** P. 7 line 15: this will be corrected in the revised paper.
    **Resulting changes:** Line 180: the sentence now reads: "…realizations that…"

- Page 15: if the grid is too sparse, then limited or no information is present which can help reconstruct missing patterns is present (repetition of "is present")
    **Author response:** P. 15 line 9: this will be corrected in the revised paper.
    **Resulting changes:** Lines 452-453: The repetition of the word present has been removed

- Page 29: "increased"
    **Author response:** P. 29 line 18: this will be corrected in the revised paper.
    **Resulting changes:** Lines 652: "increase" has been changed to "increased"

- Page 21: comparing a "realization" (no "s")
    **Author response:** P. 21 line 27: this will be corrected in the revised paper.
    **Resulting changes:** Line 292: the "s" has been removed from "realizations"

- Page 35: Journel, A. G.: "Combining Knowledge From Diverse Sources: An Alternative to Traditional Data, , 34(5), 2002". (The name of the Journal is missing, "Mathematical Geology")
    **Author response:** The mentioned reference will correctly contain the journal name in the revised paper.
    **Resulting changes:** The journal name has been added

**Response to anonymous referee #2**

**General comments:**
*'The manuscript "Contributions to uncertainty related to hydrostratigraphic modeling using Multiple-Point Statistics" presents an interesting study where the uncertainty related to the input data required by a multiple-point statistics (MPS) simulation framework is investigated. The research described in the manuscript, although focused on a specific case study in Denmark, could have a broader applicability and would probably be of interest for the HESS readers.*
*Nevertheless, I believe that the manuscript contains some major issues that should be addressed by the authors before its publication. In particular, my concerns are related to three aspects: 1) The structure of the manuscript, 2) some missing details/discussion about important aspects of the parameterization of the methodology, 3) the way mathematical relationships are expressed.'*

**RC1:** *'Manuscript structure: A number of techniques are used within the manuscript to complete the quite complex simulation framework. Some of them are used multiple times and in different contexts (for example, the tau model). Therefore, putting their description in a separate section "Methods" would be much more helpful and would help the reader in orienting himself inside a quite complex work-flow. At the moment, the description of the methods is spread all around the manuscript, sometimes together with the results, quite often with some repetition, which makes reading the manuscript not a smooth task. A clear example of this "breaking the rhythm" of the manuscript is for example at page 18. Also, here the description of the technique is made at the wrong place, because the method was already applied some step before in the work-flow. Another example is at page 21, where a 2D example is introduced to explain the EDT.*
*In addition, the comparison methods (EMR-maps,...) and the distances (EDT...) definitions would deserve a separate section, maybe just after or within the "methods" section. There are also many locations, in particular in the "Results" sections, where too many details which would be more appropriate for the "Discussion" section are anticipated (see for example pages 27-28, lines 6, 10-11).'*

**AC1:** Under the preparation of the manuscript we have worked iteratively on the structure of the manuscript and ended up with this structure as the most reader friendly consisting of a large Material and methods section (section 2), divided into the study area (2.1 The kasted study area), a general method description section (2.2 Multiple-Point Statistics (MPS) and single normal equation simulation (snesim)), a detailed description of the MPS modeling set-up with the methods related to each case (2.3 MPS modeling setup) and lastly a section of the methods used for comparing the simulation results. Generally, the style chosen for this manuscript was to explain some of the methods utilized in the, as mentioned, quite complex simulation framework was to use practical examples. This often aids the reader with figures and a purpose for applying the given method.
It does seem warranted to create a separate section for the general methods.

**Changes to the manuscript:**
The old section 2. Materials and Methods are split up in three sections: 2 The Kasted study area, 3 Methods, 4 MPS modeling setup.
Lines 145-171: The section describing the Kasted study area has been assigned to its own section, "2. The Kasted study area"
Lines 171-322: A dedicated methods section has now been added, "3. Methods", which together with the new section "4. MPS modeling setup" replaces the methods part of the old materials and methods section, "2. materials and methods section".

Lines 193-214: A sub-section describing the Tau model has been added, "3.1.1 The Tau model: Combining conditional probabilities"
Lines 215-231: A description of the direct sampling method for reconstructing incomplete datasets has been added, "3.2 Reconstructing incomplete datasets using Direct Sampling"
Lines 232-322: The old section "2.4 Comparing simulation results" has been moved to the new Methods section
Lines 325-524: The section describing the MPS setup and the simulation cases has been assigned to its own section 4. MPS modeling setup cleaned for method descriptions.

*RC2:* *'Section "Basic modeling set-up": This section is somehow quite confusing, because the authors mix the description of the "Basic modeling set-up" with the Egebjerg TI description. I suggest to better separate the description of the various cases.'*

**AC2:** This was a mistake. The reference to Table 2 for some reason included text from the section above or below Table 2.

**Changes to the manuscript:**
Line 406: The reference to Table 2 has now been fixed and the Egebjerg TI is not mentioned under section "4.1 Basic modeling setup"

**RC3:** *'Tau model usage: The tau model represents one of the crucial steps of the methodology, because it is used to take into account the soft constraints provided by geophysics, but also to combine the "borehole probability" with the SkyTEM one (Fig.7). Although some information about the tau weights are provided (i.e., in appendix), I would suggest to discuss at least briefly their choice. For example, many of the considerations made by the authors would be strongly influenced by the choice of the tau weights (see for example line 32, page 30). Some insights about the choice of these weights are provided by Allard et al (2012, DOI: 10.1007/s11004-012-9396-3). Also, what happens when the weights are_1? (see pp18, equation 2).'*

**AC3:** Here we refer to the Author Comment 4 (AC4) in the response to anonymous referee #1. Briefly, in relation to the choice of Tau parameters for the Tau model it should be mentioned that a series of tests have actually been carried out to select the final Tau values.

**Changes to the manuscript:**
Lines 508-511: The exhaustive tests for choosing the tau model parameters are described

**RC4:** *'Case studies labelling: The provided table that summarizes all the case studies is of course useful, but overall into the manuscript (for example, in figure captions), there is very often a redundancy and some of the details of the different methods, which are repeated multiple times.*
*Maybe you should reference much more often to Table 1 and to the "codes" like "Case 1a", "Case 1b" only, and avoid repeating the detailed differences. One example of these repetition can be observed in Figure captions (see for example Fig.9, page 25).'*

**AC4:** This would certainly make the paper more concise in certain parts. However, it would also require the reader to constantly avert his/her attention to Table 1.

**Changes to the manuscript:**
Lines 578-581: The figure text related to Figure 9 has now been simplified and a reference to Table 1 is added

**RC5:** '*Introduction pp4, lines 20-XX: Here I would also mention the problems related to the solution of the inverse problem (IP) in itself. By the way, this also somehow motivates your efforts in trying two different inversion techniques, like SCI and sSCI.*'

**AC5:** This is a good point.

**Changes to the manuscript:**
Lines 111-119: A brief description of the problems related to the inversion process have been added to the introduction

**RC6:** '*pp5, lines 27-: Here I believe you are already providing too much details for an introduction.*'

**AC6:** Okay, we will provide less detail

**Changes to the manuscript:**
Line 136-144: Less detail is now provided in the mentioned paragraph

**RC7:** '*Mathematical formulation: The mathematical formulation is often cumbersome, because very often long text lines are used to define quantities and as subscripts. I strongly suggest to lighten the notation avoiding long text lines, and using the many letters provided by the alphabets. For example, N and M could be used instead of Nrealizations and Ncells; another example is the definition of Di;j (page 22). In addition, some relationship could be condensed and generalized. In this way, they could be written only once and contribute to shorten the manuscript. See for example (7) and (8) at page 21.*'

**AC7:** We agree that the mathematical formulations could be written in a more concise manner. However, regarding eq. (7) and (8), again, we believe it aids the reader to first see the formula with a specific case in mind and then generalize afterwards.

**Changes to the manuscript:**
Line 253: Equation has been simplified: "$N_{reals}$" changed to "N",
Line 285: Equation has been simplified: "cog.model" has been changed to "$m_{cog}$", "realization$_i$" has been changed to "$m_{r,i}$", the expression "$d_{EDT}^{cog.model}$" has been changed to "$d_{EDT}^{m_{cog}}$", and the expression "$d_{EDT}^{realization_i}$" has been changed to "$d_{EDT}^{m_{r,i}}$"
Line 294: The equation has been simplified: the expressions "model$_A$" and "model$_B$" have been simplified to "$m_A$" and "$m_B$"
Line 312: The equation has been simplified: the expressions "realization$_i$" and "realization$_j$" have been changed to "$m_{r,i}$" and "$m_{r,j}$"

**RC8:** *'TI non-stationarity: In section 3.1 but also in other parts of the manuscript the imprecision of MPS in reproducing some features is clearly depicted. However, I believe that many of the encountered problems are due to the non-stationarity of the used TI. Therefore, although of course taking into account for the geophysics helps, I would suggest to at least mention and briefly describe the role of the non-stationarity of the TIs.'*

**AC8:** The geophysical data is so spatially dense that TI non-stationarity is not as big of an issue as when less conditioning data is present. We therefore decided to not include this complication, to not confuse the reader. However, it might be wise to mention that this is the case and that even though the Tis are non-stationary, the spatially dense geophysical data actually helps in the matter.

**Changes to the manuscript:**
Lines 697-701: The non-stationarity of the Kasted model and Tis are discussed in relation to the density and overall coverage of the geophysical dataset

**RC9:** *'Variograms: Could you briefly mention which variogram model you used for example to create the borehole footprint (pp18, lines 3-6)?'*

**AC9:** This is mentioned both in the text where you indicated, but also in the Appendix under section "A1".

**Changes to the manuscript:**
Lines 890-891: the variogram model type (exponential) is now mentioned under Appendix "A1"

**RC10:** *'Fig.4: Please add a comment related to the spatial scale of the Egebjerg TI, which is quite different from the other two. Also, it would be quite nice to add a sub-figure containing the same vertical proportions for the borehole logs.'*

**AC10:** That is a good point! A comment related to the spatial scale of the Egebjerg TI will be added. A subfigure containing the vertical proportions of the borehole logs was not added since the available boreholes were largely drilled for the purpose of groundwater exploration and management. This means they are biased towards more permeable hydrostratigraphic layers such as coarse sands and gravels, and impermeable clay layers are generally underrepresented.

**Changes to the manuscript:**
Line 420: A comment has been added commenting on the fact that the Egebjerg TI is larger than the other TIs.

**RC11:** *'Fig.11: The label "Realization number" in the vertical axis of part A is too close to part B and is therefore misleading. Also, I believe that the results of part B could be condensed using box-plots, one box-plot for each case. In this way, the fictitious and misleading order of the "realization number" would be by-passed.'*

**AC11:** Yes, the Fig.11B of the figure will be given a bit more space so that it is clear that the "Realization number" is part of Fig.11A. We would prefer to not use boxplots since they mainly provide summary statistics and we would prefer that the reader can see the actual distance values instead.

**Changes to the manuscript:**
Lines 632-634: Sub-figure B has now been moved so that sub-figure A and B are above and below each other

**Technical comments:**
- pp6, line 9: Please check the order in "33 line km spatially…"

  **Author response:** This will be re-ordered correctly.
  **Resulting changes:** Lines 160-161: The sentence has been revised and re-ordered.

- pp8, line 30: It looks like the reference to Fig.4 is missing between Fig.3 and Fig.5.

  **Author response:** We will add a sentence with reference to Fig. 4, so it will be mentioned before Fig. 5
  **Resulting changes:** The figure numbers have changed, but the figures are now mentioned in the correct order in the text.

- pp9, line 5: "data is" => "data are"

  **Author response:** This will be corrected in the revised paper.
  **Resulting changes:** Line 399: The sentence now read "...data are..."

- pp18, line 18: "[2,1]" is somehow confusing with the index that you introduce some equations before... I would specify that they are float values, writing explicitly 2.0 or 1.0.

  **Author response:** This will be added to the revised manuscript.
  **Resulting changes:** Line 535: the values have now been changed to floating point values ´, *i.e.* [2.0,1.0]

- pp20, line 25: Maybe "(3)" => "(4)"?

  **Author response:** This will be corrected.
  **Resulting changes:** Line 272: The equation references have now been fixed

- pp21, equation (6): Please check for the missing i subscript to v

  **Author response:** There is no missing subscript. The "**u**" symbol represents a single location vector, while the "**v**" is a set of vectors which are a subset of "V". Therefore, the formula only shows the computation of the $d_{EDT}$ at a single location. The process must be repeated for all points for the grid. Perhaps this should be mentioned in the revised paper.
  **Resulting changes:** None

- pp21, line 20: Here Delta appears in the formula, but not in the following text... please check.

  **Author response:** This will be corrected in the revised paper.
  **Resulting changes:** Delta has been added to the following lines: 302, 304, 305, 306, 307, 320, 321 and 326

- pp22, line17: "realizations using" => "realizations computed using" (?)

  **Author response:** This will be corrected in the revised paper.

**Resulting changes:** Line 328: Changed to: "...realizations computed using..."

- pp27, lines 6-11, 30-31: This is somehow repetitive. Please try to avoid repetitions also in other locations in the text, but in particular in this section.

    **Author response:** We couldn't find the particular repetition mentioned here, but will try to remove as many of such repetitions in the revised paper.

    **Resulting changes:** None

- pp30, line 26: "to alter...?"

    **Author response:** Will be corrected, however, it seems like it is a logical statement.

    **Resulting changes:** line 708: Text has been modified accordingly

**Response to anonymous referee #3**

**General comments:**
*'This paper is the product of a nice piece of work and is very interesting. My main concerns are related to the introduction, where some transitions and justifications are missing and to the methodological section, that goes too fast into details. I therefore suggest the following minor revisions.'*

**RC1:** *'In the introduction, transitions are often missing between paragraphs, for instance, page 2, between lines 14 and 15. It is a bit jumps from one topic to another. Related to the paragraph comprised between lines 3 and 14, you might cite the following paper that discuss uncertainty and bias in training images : Ferré, Ty. "Revisiting the Relationship Between Data, Models, and Decision-Making." Groundwater 55, no. 5 (2017): 604-614. Pirot, Guillaume. "Using training images to build model ensembles with structural variability." Groundwater 55, no. 5 (2017): 656-659.'*

**AC1:** Good points.

**Changes to the manuscript:**
The introduction has been cleaned up so that transitions between paragraphs are less abrupt in the revised paper.
Lines 89-101: The paragraph describing the overall goal of the paper has been moved up
Line 84: The papers mentioned above by Ty Ferré and Guillaume Pirot are now cited.

**RC2:** *'General justifications are given in the first paragraph of the introduction, but the authors should also justify why they chose this specific sites, and what they bring or want to improve, with regards to previous studies conducted at the Kasted site.'*

**AC2:** That is a good point. The main justification for using the Kasted site was that the geology of the area was fairly simple, thus the dataset is also, to a degree, simple. Furthermore, a geological model of the area had already been compiled, meaning we already had information about the hydrogeology.

**Changes to the manuscript:**
Lines 98-100: A reflection of what we hope to improve in regards to the Kasted model has been added
Lines 154-159: The description of the Kasted survey area has been expanded and a general justification for the usage of the Kasted data has been added

**RC3:** *'You should also define your notions of hard and soft conditioning clearly in the introduction'*

**AC3:** Okay.

**Changes to the manuscript:**
Lines 102-105: Our notions of hard and soft conditioning has been added

**RC4:** *'Then, in section 2, when presenting the study area, the historic of previous research could be explained/clarified. Page 6, line 5: one 'complex' too much? Page 6, line 14: how is defined the 'selected quality threshold?".'*

**AC4:** If anonymous referee #3 is referring to the fact that we should explain previous research in the Kasted area, then the main research which has been conducted in the area is compiling the cognitive 3D geological model of the area. Therefore, we do not desire going into too much detail.

**Changes to the manuscript:**
Lines 154- 159: Additional information regarding the Kasted area has been added, introducing the essential feature known as tunnel valleys.
Line 150: The poor usage of the word 'complex' is corrected
Lines 163-165: References to the borehole quality assessment scheme we utilized has been added and the 'selected quality threshold' is elaborated upon

**RC5:** *'The main aspect, regarding the method subsection is that the reader is lost in details from the beginning. A big picture of the approach is missing. In the way of presenting, I would recommend to give first an overview of the method, and progressively go into details.'*

**AC5:** Perhaps it would be wise to include an overview of what we hope to achieve by stochastic modelling.

**Changes to the manuscript:**
The structure of the paper has been changed significantly. A new Methods section has been added which contains a generalized description of the methods before they are invoked, which clearly separates the theory behind the methods from how it is used. Furthermore, the description of the Kasted area has been given its own section. The MPS modeling setup section has got an introductory section.

**RC6:** *'Page 7, lines 7 to 15 are not very clear. Can you reformulate?'*

**AC6:** Will be reformulated in the revised paper.

**Changes to the manuscript:**
Lines 175-182: The paragraph has been reformulated

**RC7:** *'Page 8, line 26, figure 5 is called, while figure 4 has not been called.'*

**AC7:** We will fix this by adding a reference to Figure 4 before we call Figure 5.

**Changes to the manuscript:**
The figures are now invoked in the correct order

**Miscellanous Corrections:**

- The author list has been updated so that co-author Anne-Sophie Høyer is on the list
- Lines 183-185: References have been updated so that (Høyer et al. 2017) is now correctly categorized as a hydrology paper
- Line 268:  A reference to the new tau model section has been added
- Line 210: The denominator was missing a "P" in the last formula ($x_2$=…)
- All figures have been updated with a larger sub-figure label
- Lines 422-424: Figure 5 text has been updated so that the usage of "slice" and "cross-section" are correct
- Lines 483-487: Figure 7 text has been updated so that the usage of "slice" and "cross-section" are correct
- Lines 506-507: A reference to Appendix A1 is has been added
- Lines 658-660: Figure 8 text has been updated so that the usage of "slice" and "cross-section" are correct
- Lines 582-586: The text has been shortened and simplified and a reference to table 1 has been added
- Line 602: The Barfod *et al.* (2018) reference has been updated since it is now a published paper
- Lines 677-683: The paragraph has been edited to make it more readable
- Line 774: "I" has been replaced with "We"

**Marked-up manuscript**

[revised manuscript text omitted]

---

## Author Response (AR2)

Dear Mr. Carrera,

Thank you for your interest and work with improving the manuscript. We really appreciate the input, and the fact that you can see the relevance and potential of the research contained in the paper.

We have decided to revise the paper according to as many of your comments as possible, since this will help to improve the paper even more.

Best,
Adrian S. Barfod

**Response to editor comments, hess-2017-734**

Firstly, the authors would like to thank the editor, Jesus Carrera, for taking his time to read the manuscript and providing detailed and constructive comments. The comments, questions and suggestions are addressed in the following response.

**\*EC: Editor Comments**
**\*AC:** Author Comments

*NOTE: All line numbers in this document refer to the revised paper and not the "manuscript changes document" or any previously submitted versions of this paper.*

**Response to editor comments**

**General comments:**
'Dear authors,

The paper is good. You were lucky in having three thorough positive reviewers and you have responded in kind, leading to an improved paper. Moreover, the topic you address is important (I have often wondered about many of the questions you respond). I am left with many questions, but this may reflect the complexity of your goal. Therefore, I conclude that it has a great potential (although you never know whether a paper is going to have a large impact, and while uncertainty is recognized as important by all, few actually care much; I'd recommend you to present your results in congresses). I share below with you some of the possible improvements that come to my mind after reading the final version (the list is long, but most comments are very easy to handle).

Do not take these comments as compulsory in any way, but rather as an invitation to have a final thorough reading while feeling free to write your thoughts out and hopefully transform a good paper into an excellent one (but, please, if you change things, let me know what you have changed to facilitate my life!).

Excellent job!

Jesus.'

**EC1:** *Line 56 "only 410 boreholes" sounds weird for most of us (I have never had 410 borehole descriptions!)… especially considering that in the abstract you do not mention the spatial extent of the study area. When I realize that the surface area is 45 km2, or some 10 "good" boreholes per km2, I conclude that the density is exceptionally high. While it is clear that the geological complexity is such that a deterministic description is not possible, I still believe that "only 410 boreholes" is inappropriate.*
The fact that "The borehole lithology logs infer local changes in the immediate vicinity of the boreholes" depends on the geological setting, which you have not yet described (in the end, I believe the statement is true in your case).

**AC1:** That is a good point. We do generally have a high borehole density in Denmark compared to other places in the world and often forget this fact.

**Changes to the manuscript:**
Line 57: The word "only" has now been dropped.
Lines 58-59: The sentence has been modified so that it is clear that we are talking about areas with a high degree of geological heterogeneity.

**EC2:** *Line 59: "Finally, the importance of the TI was studied. An example was presented where an alternative geological model from a neighboring area was used to simulate hydrostratigraphic models" does not read well (it is not clear to me what the "finally" refers to). Perhaps better "The importance of the TI was studied. An example was presented …" or "The importance of the TI was studied through an example where an alternative geological model from a neighboring area was used to simulate hydrostratigraphic models'*

**AC2:** Agreed, revision has been carried out.

**Changes to the manuscript:**
Line 60: The word "finally" has now been dropped and the word "also" has been added so the sentence now reads: "The importance of the TI was also studied."

**EC3:** *'Line 77-78. The statement "E.g. indicating different levels of uncertainty in different subparts of the model domain" should perhaps be in parentheses (e.g., indicating different levels of uncertainty in different subparts of the model domain).'*

**AC3:** Agreed.

**Changes to the manuscript:**
Line 79-80: The sentence has been put in a parenthesis as suggested.

**EC4:** *'Line 83. My standard references for MPG (I prefer MPG to MPS, which suggest discrete multivariate distributions) are those of Strebelle (2002) and Mariethoz and Renard (2010). You cite both in the text later, but I find surprising your choice here. The first time I read your paper I had to go to the references to ascertain that you were indeed talking about what I call MPG.'*

**AC4:** We are not referring to articles which present the MPS/MPG methods themselves, but rather studies where MPS/MPG has been used to create stochastic models of the subsurface. We will specify this in the text

**Changes to the manuscript:**
Line 85-86: added: "where ensembles of models are produced,"

**EC5:** *'Lines 88-100: The structure of a paper is always arguable, but I believe that this paragraph belongs better in the "methods" section than in the introduction. Usually, the introduction is used for revising the state of the*

*art, so as to motivate the objectives of the paper. In your case, the motivation and objectives are clear and well established, but this paragraph looks awkward in the introduction as you discuss methods and the site, both of which are described later. I realize that this type of change is tedious and may introduce other "losses" of logic, but I still invite you to consider revising the structure of the introduction and methods. The same can be said about line 111, or lines 121-122. In all these cases, you identify sources of uncertainty and say how you are going to address it. The end result is that, after reading the introduction, the reader is left with a mixture of loss of confidence in geological modeling and a somewhat vague feeling that you are going to somehow assess them. I invite you to consider revising all sources of uncertainty to conclude that the objective of the paper is to assess their impact on the uncertainty of the geological model and, then, describe in a compact manner how you are going to do it (perhaps in an itemized way).'*

**AC5:** We agree that this might improve the manuscript and will therefore try to accommodate the issues mentioned in relation to the introduction.

**Changes to the manuscript:**
Line 115: The statement which refers to the presentation of both cases, with and without reconstructing the geophysical data, has been removed.
Line 124: The statement describing the specific method of dividing the boreholes into quality groups has been removed.
Lines 131-137: The paragraph describing the overall goal of the study/research has been removed and parts of it has been re-used for compiling a paragraph describing the goals of the study in a more general way, without discussing the specific methods or the Kasted site.

**EC6:** Line 104: "provide" instead of "provides"

**AC6:** Agreed.

**Changes to the manuscript:**
Line 111: "provide" now replaces the word "provides"

**EC7:** Line 105: "Resolution decreases with depth, and diminishes at a specific depth, which is dependent on the geophysical method" reads weird. Perhaps better "Resolution decreases with depth, specially beyond a specific depth, which is dependent on the geophysical method"

**AC7:** The sentence should be corrected.

**Changes to the manuscript:**
Lines 109-110: The sentence now reads "Resolution decreases with depth, especially beyond a specific depth, which is dependent on the geophysical method." as suggested.

**EC8:** Line 153: I do not know what you mean by "tunnel valleys"

**AC8:** The term "tunnel valley" is synonymous with buried valley, however, we do agree that it is simpler to just use the same term throughout the paper.

**Changes to the manuscript:**
Line 147: The usage of the word "tunnel valley" has now been replaced by the word "buried valley"

**EC9:** Figure 1: Readers that do not know SKYTEM may be confused about Figure 1C. I assume that the blue lines are the trajectories of SkyTEM flights, you may want to say so (instead of soundings) in the caption and identify them with a blue line (instead of a dot) in the legend.

**AC9:** The blue "lines" are actually the sounding locations (dots) which are spaced so close that it looks like a line, therefore, since the legend represents the actual sounding locations, we believe that this is the most accurate way of portraying the data, though we see you point and  add a note on it in the figure caption

**Changes to the manuscript:**

Line 164-165: The following text was added: *"Note, that the SkyTEM soundings are sampled so dense that the dots marking each individual sounding merge into blue lines.**"**

**EC10:** Line 176: I would not say "The TI thus provides a conceptual geological understanding". If by conceptual understanding, one means a qualitative description of the site and its origins, the TI provides neither. To me, the "conceptual geological understanding" is what you provide in lines 147-150. Although the TI can be based on geological understanding (and, as such, it is indeed a conceptual statement), the TI provides simply a quantitative description of the expected (often only small scale) spatial variability patterns of the random function of interest (in your case textural description).

**AC10:** The conceptual geology can also be thought of as the specific geological patterns/architecture which we wish to represent in our hydrostratigraphic model, e.g. buried valleys. If the TI contains a series of buried valleys then the statistics and quantitative description contained in the TI describes the conceptual geology.

**Changes to the manuscript:**
The comment didn't result in any changes.

**EC11:** Section 3: I realize that you have made an effort at improving the structure of Section 3, and I think that further improvement is possible (structure is important for Cartesian people like me and, I assume, most of the readers understanding section 3). Specifically, snesim and DS are two methods for doing MPS, while the tau approach is a method to combine conditional probabilities. Therefore, including the tau-model as a subsection of snesim is misleading. You may choose: 3.1 MPS; 3.1.1 snesim; 3.1.2 DS; 3.2 tau model. You may also choose: 3.1 MPS: snesim; 3.2 MPS:  DS; 3.3 tau model. The latter may be better for your goal because you appear to have chosen (for reasons that are not clear to me to use DS and snesim for different goals).

**AC11:** That is a very good point, the manuscript will be changes accordingly.

**Changes to the manuscript:**
Section 3: The section has been changed according to option 1: 3.1 MPS; 3.1.1 snesim; 3.1.2 DS; 3.2 Tau model

**EC12:** The tau model: the description of the tau model can also be improved. The one you make is somewhat abstract (Journel style), which may be appropriate for a statistical journal but should be made clearer for a hydrological journal: Possible improvements include:
- The tau value can be used to qualify the degree of redundancy among data sets (see www.ccgalberta.com/ccgresources/.../2003-130-naivebayes.pdf)
- Use a bold face for Di data events (as used in eq. 1). Also, explain what data events are. At first, I thought it referred to actual data sets (e.g., core logs), but later it appears to refer to probability grids (a concept you may define as well)
- In line 210, you should define x_i as distances from a given realization of A to its true value.
- Line 212: the statement "where the tau values are assigned as follows: [τ1,τ2]" does not really describe how tau values are assigned.

**AC12:** We would like to keep the abstract description if possible, since this describes the actual equations used in this particular study.

The issue with bold face letters representing the has been fixed many times but Word keeps changing the equations to being bold face. We should be able to handle this during publication.

Defining that x_i as the distance from a given realization of A to its true value seems very confusing since the term distance in this case refers to the inverse of the probability as opposed to the Euclidean distances defined in the rest of the paper.

**Changes to the manuscript:**
Lines 213-214: The definition of a data event has been added
Line 225: The sentence now reads: "where the tau values are: $[\tau_1,\tau_2]$".

**EC13:** Line 259: K was already defined in line 248 (you do not need to define it again).

**AC13:** Good point, we would like to avoid such repetitions

**Changes to the manuscript:**
Line 255: The definition of K has been removed.

**EC14:** Line 276: You do not need the comma after (2003).

**AC14:** Okay

**Changes to the manuscript:**
Line 271: The comma has now been dropped

**EC15:** Consider addressing comment 2 of reviewer 1 (actually your response) in the manuscript.

**AC15:** The comment is a good discussion point and has been added to the discussion.

**Changes to the manuscript:**

Lines 696-705: A paragraph discussing the presence of small-scale patterns in the resulting realizations has been added